# Dual targeting of polyamine synthesis and uptake in diffuse intrinsic pontine gliomas

Aaminah Khan[1], Laura D. Gamble [1], Dannielle H. Upton [1], Caitlin Ung [1], Denise M. T. Yu[1], Anahid Ehteda [1], Ruby Pandher[1], Chelsea Mayoh [1], Steven Hébert [2], Nada Jabado [3], Claudia L. Kleinman [2], Mark R. Burns [4], Murray D. Norris[1,5], Michelle Haber[1,5], Maria Tsoli [1,7] & David S. Ziegler [1,6,7 ✉]

Diffuse intrinsic pontine glioma (DIPG) is an incurable malignant childhood brain tumor, with no active systemic therapies and a 5-year survival of less than 1%. Polyamines are small organic polycations that are essential for DNA replication, translation and cell proliferation. Ornithine decarboxylase 1 (ODC1), the rate-limiting enzyme in polyamine synthesis, is irreversibly inhibited by difluoromethylornithine (DFMO). Herein we show that polyamine synthesis is upregulated in DIPG, leading to sensitivity to DFMO. DIPG cells compensate for ODC1 inhibition by upregulation of the polyamine transporter SLC3A2. Treatment with the polyamine transporter inhibitor AMXT 1501 reduces uptake of polyamines in DIPG cells, and co-administration of AMXT 1501 and DFMO leads to potent in vitro activity, and significant extension of survival in three aggressive DIPG orthotopic animal models. Collectively, these results demonstrate the potential of dual targeting of polyamine synthesis and uptake as a therapeutic strategy for incurable DIPG.

[1] Children's Cancer Institute, Lowy Cancer Research Centre, UNSW Sydney, Kensington, NSW 2052, Australia. [2] Lady Davis Institute for Medical Research, Jewish General Hospital, Department of Human Genetics, McGill University, 3999 Côte Ste-Catherine Road, Montreal, QC H4A 3J1, Canada. [3] Department of Pediatrics, McGill University Health Center, 1001 Decarie Boulevard, Montreal, QC H4A 3J1, Canada. [4] Aminex Therapeutics Inc., Suite #364, 6947 Coal Creek Parkway SE, Newcastle, WA 98059, USA. [5] Centre for Childhood Cancer Research, UNSW Sydney, Kensington, NSW 2052, Australia. [6] Kids Cancer Centre, Sydney Children's Hospital, High St, Randwick 2031, Australia. [7] These authors contributed equally: Maria Tsoli, David S. Ziegler. ✉email: dziegler@unsw.edu.au

Diffuse intrinsic pontine glioma (DIPG) is an extremely aggressive brainstem tumor with a median survival of less than one year, and the most common form of high-grade glioma in children[1–4]. With no effective treatments, DIPG is uniformly fatal and remains the leading cause of brain tumor-related death in childhood[5]. Despite over 250 clinical trials with different therapeutic agents there has been no improvement in the dismal prognosis of DIPG[6] with focal irradiation still remaining the standard treatment. Over the past decade, the implementation of autopsy and biopsy sampling has increased our understanding of the molecular landscape of DIPG tumors[7,8]. Nearly 80% of DIPG tumors harbor an H3K27M mutation leading to loss of H3K27 trimethylation and aberrant gene expression. Due to the significant impact of H3K27M on DIPG growth and patient outcome, DIPG has been reclassified by the WHO into a distinct category of Diffuse Midline Gliomas with K27M mutation (DMG) which also consists of thalamic, brainstem, and spinal tumors[9]. Furthermore, the development of patient-derived cultures and orthotopic models has allowed the preclinical testing of a variety of epigenetic, targeted agents, and immunotherapies[4,10–12]. However, the activity of most therapeutic compounds tested in vivo has been limited, akin to the highly treatment resistant tumors seen clinically. The development of novel therapies which display low toxicity and ability to cross the blood–brain barrier will provide new therapeutic options for DIPG patients.

Polyamines have been investigated in aggressive cancers as they play a pivotal role in multiple cellular processes and facilitate rapid cell proliferation[13,14]. Polyamine intracellular concentration is tightly regulated through biosynthetic and catabolic pathways, as well as uptake of polyamines from the microenvironment (Supplementary Fig. 1)[15,16]. ODC1 is responsible for the decarboxylation of ornithine into putrescine, the first polyamine in the biosynthetic pathway[17–20]. ODC1 activity is frequently elevated in cancer[21] and inhibiting ODC1 with the small molecule irreversible inhibitor difluoromethylornithine (DFMO) has shown activity in various adult cancer models[17,18,22,23]. However clinical results in adult cancers have been limited[24,25], at least in part due to the activation of compensatory mechanisms including an increase in polyamine uptake from the microenvironment[26,27]. Dual targeting of polyamine synthesis and uptake has shown promise as a more potent method of targeting the polyamine pathway in neuroblastoma—an epigenetically driven childhood cancer[28,29].

In this study, we demonstrate the anticancer potential of polyamine pathway inhibitors in preclinical models of DIPG. Our results show that the polyamine pathway is upregulated in DIPG. The combination of polyamine synthesis inhibitor DFMO and transport inhibitor AMXT 1501 leads to significant depletion of polyamine levels leading to reduction of cell proliferation, clonogenic potential and cell migration, along with the induction of apoptosis. The combination of DFMO with AMXT 1501 enhances significantly the survival of three orthotopic models of DIPG and can be combined effectively with irradiation. An adult clinical trial is currently underway for the combination of DFMO with AMXT 1501 (NCT03536728). These preclinical results will pave the way for the development of a pediatric clinical trial for patients with DIPG.

## Results

### Polyamine pathway gene expression and inhibition in DIPG.
To identify the role of the polyamine pathway in DIPG, we examined gene expression of the key regulators of this pathway in DIPG/DMG tumors ($n = 45$) compared with normal fetal brain samples ($n = 11$). Data were obtained from the Zero Childhood Cancer (ZCC) precision medicine platform and McGill University[30]. ODC1 was significantly overexpressed, as were all biosynthetic genes of the polyamine pathway compared to the normal brain (Fig. 1a, Supplementary Fig. 2a, Supplementary Table 1). ODC1 over-expression was independent of H3K27 mutation status (Supplementary Fig. 3). Correspondingly, decreased expression of all negative regulators of polyamine production (including the rate-limiting enzyme spermine/spermidine $N_1$-acetyltransferase (SAT1) that drives polyamine catabolism) was found in the same cohort (Fig. 1b, Supplementary Fig. 2b, Supplementary Table 2). Similarly, ODC1 protein and mRNA levels were increased in a panel of patient-derived H3K27M DIPG cell cultures compared with three normal human astrocyte cultures (NHA, P000302, RA038), while SAT1 mRNA levels were decreased (Fig. 1c–e, Supplementary Fig. 4). Overall, a moderate negative correlation was found between the IC50 values and ODC1 protein levels (Supplementary Fig. 5). To determine whether higher levels of polyamine synthetic enzymes were associated with increased polyamine levels in vivo, we measured the levels of putrescine, a polyamine synthesized directly from ODC1, in the brains of xenografted animals. We observed that the brainstem region of nude mice orthotopically injected with patient-derived HSJD-DIPG007 cells had higher levels of putrescine than the brainstem of control mice (Fig. 1f) and consequently low spermidine to spermine ratios (spd:spm) (Fig. 1g). Furthermore, when HSJD-DIPG007 cells were supplemented with exogenous polyamines in vitro, the cells displayed both increased cell growth and migration (Fig. 1h and Supplementary Fig. 6). Collectively these results support the rationale that targeting the polyamine pathway as a treatment strategy for DIPG. In addition, treatment of primary DIPG cell lines with DFMO led to decreased ODC1 expression and reduced the proliferation of primary DIPG cells, with minimal effect on normal healthy astrocytes (NHA) and MRC-5 fetal lung fibroblast cells (Fig. 1i and Supplementary Fig. 7).

### Polyamine transport gene expression and inhibition in DIPG.
While DFMO shows anti-DIPG activity, one method by which cancer cells can compensate for the effect of ODC1 inhibition on polyamine synthesis is by increasing uptake of polyamines via polyamine transporters[29,31,32]. We have previously shown that SLC3A2, which forms the heavy chain of the CD98 glycoprotein, is a critical membrane transporter for polyamine uptake in neuroblastoma[29]. Using the same RNAsequencing data from ZCC and McGill University we assessed SLC3A2 expression in DIPG and found significantly higher SLC3A2 expression in DIPG tumor samples ($n = 45$) compared to fetal-derived normal brain tissue ($n = 11$) (Fig. 2a, Supplementary Table 3). Evaluating SLC3A2 expression from the ZCC precision medicine platform, we also found significantly higher expression in the cohort of DMGs compared to other high-risk pediatric cancers including relapsed/refractory high-risk neuroblastoma (Fig. 2b, c, Supplementary Table 4). Similarly to DIPG we also found higher SLC3A2 expression in HGGs comparing to other brain tumors and high-risk neuroblastoma (Supplementary Fig. 8, Supplementary Table 4).

We next sought to determine whether DIPG cells compensate for treatment with DFMO by increasing polyamine transport. We found that treatment of the HSJD-DIPG007 cells with DFMO led to a significant increase in both SLC3A2 protein and mRNA expression in a time-dependent manner (12–72 h) (Fig. 2d, e, Supplementary Fig. 9), resulting in a significant increase in uptake of radiolabelled spermidine in HSJD-DIPG007 and SU-DIPGVI cells (Fig. 2f, Supplementary Fig. 10a). Since DIPG cells are able to compensate for the effects of treatment with DFMO by

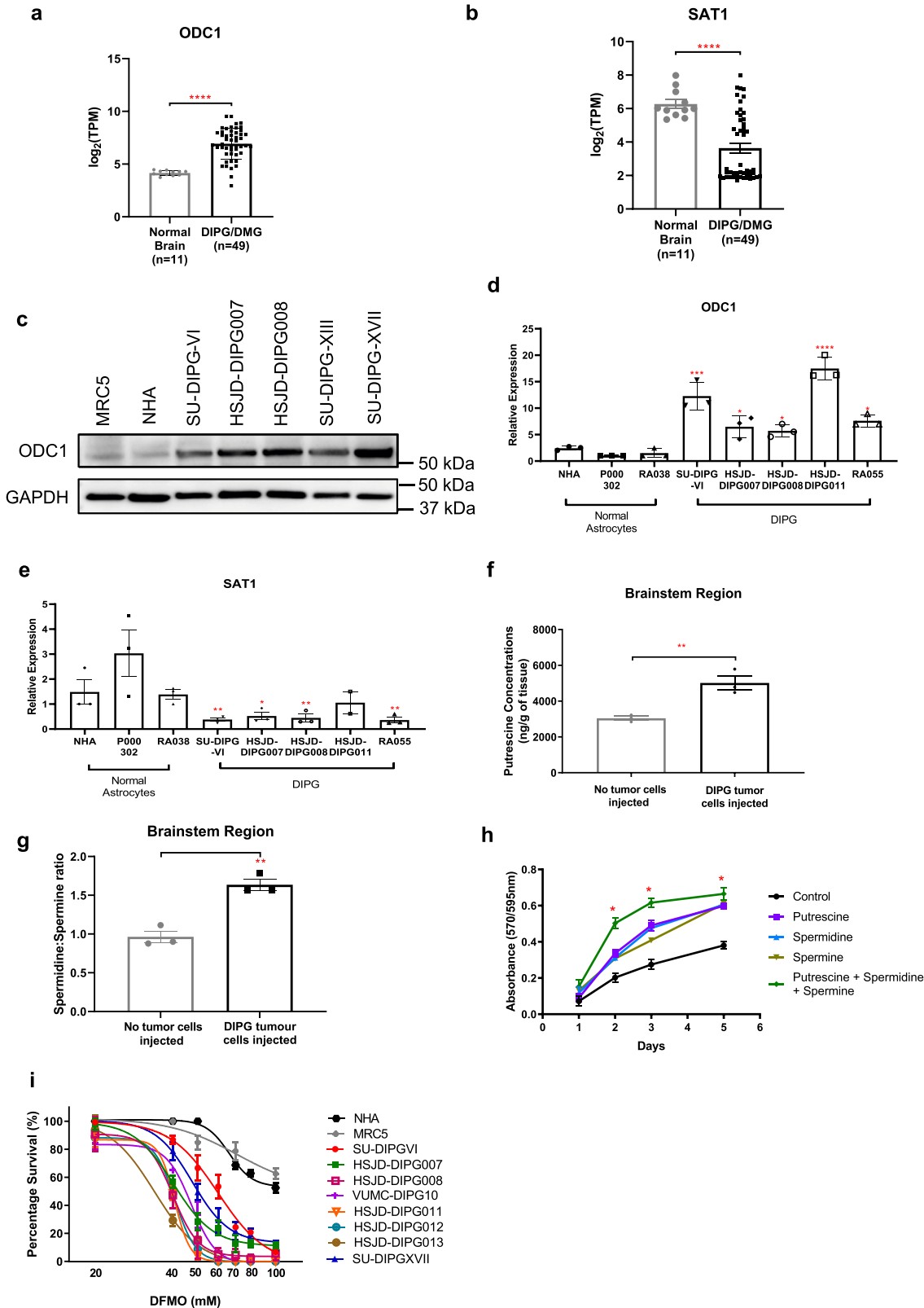

increasing polyamine uptake, we next used the transport inhibitor AMXT 1501 to test the effect of blocking polyamine transport[33]. Treatment with AMXT 1501 potently decreased radiolabelled spermidine uptake in both HSJD-DIPG007 (Fig. 2g) and SU-DIPGVI cells (Supplementary Fig. 10b), while administration of AMXT 1501 as a monotherapy reduced the proliferation of DIPG cell lines, with IC50s ranging from 4–10 μM, with minimal effect

on healthy NHA and MRC5 cells at similar concentrations (Fig. 2h). These results suggest that targeting SLC3A2 in DIPG may overcome a potential resistance mechanism to treatment with DFMO. Conversely, DIPG cells may compensate for the inhibition of polyamine transport by increasing synthesis. We found that treatment with AMXT 1501 led to increased ODC1 expression in 3 primary DIPG cultures, which was mitigated by

**Fig. 1 Polyamine synthesis and catabolic genes in pediatric brain tumors.** Expression of polyamine (**a**) biosynthetic ($p < 0.0001$) and (**b**) catabolic enzyme genes ($p < 0.0001$) in DIPG ($n = 49$) compared to normal fetal brain ($n = 11$). Data were obtained from ZCC/ PRISM Clinical trial and McGill University[30]. Protein (**c**) and gene (**d**) expression of the polyamine biosynthetic gene ODC1, and catabolic gene SAT1 (**e**) in patient-derived DIPG cell lines. Western blot is representative of two independent experiments. Putrescine concentrations (**f**) ($p = 0.0087$) and spermidine to spermine ratio (spd:spm) (**g**) ($p = 0.003$) in DIPG tumor injected brainstem region of mouse brain. **h** Exogenous polyamines were added to HSJD-DIPG007 cells at 10 μM. $p$-value was calculated by t-tests for control and addition of three polyamines at each time point. Day 2: $p = 0.00171$. Day 3: $p = 0.0115$. Day 5: $p = 0.0194$. (**i**) Patient-derived DIPG cell lines are sensitive to polyamine synthesis inhibition via DFMO, compared to normal healthy astrocytes (NHA) and normal lung fibroblast (MRC5) cells. Data are presented as mean values ± SEM of three independent experiments. *$p < 0.05$, **$p < 0.01$, ***$p < 0.001$, ****$p < 0.0001$. **a**, **b**, **f–h** Statistical analysis was performed by two-tailed $t$ tests for normal and DIPG samples. **d**, **e** Statistical analysis was performed by one-way ANOVA between normal NHA cells and patient-derived DIPG cell lines. $n = 1$ from three independent experiments. **d** NHA vs SU-DIPGVI: $p < 0.0001$, NHA vs HSJD-DIPG007: $p = 0.0478$, NHA vs HSJD-DIPG008: $p = 0.0239$, NHA vs HSJD-DIPG011: $p < 0.0001$, NHA vs RA055: $p = 0.0165$. **e** NHA vs SU-DIPGVI: $p = 0.0070$, NHA vs HSJD-DIPG007: $p = 0.0111$, NHA vs HSJD-DIPG008: $p = 0.0087$, NHA vs RA055: $p = 0.0066$.

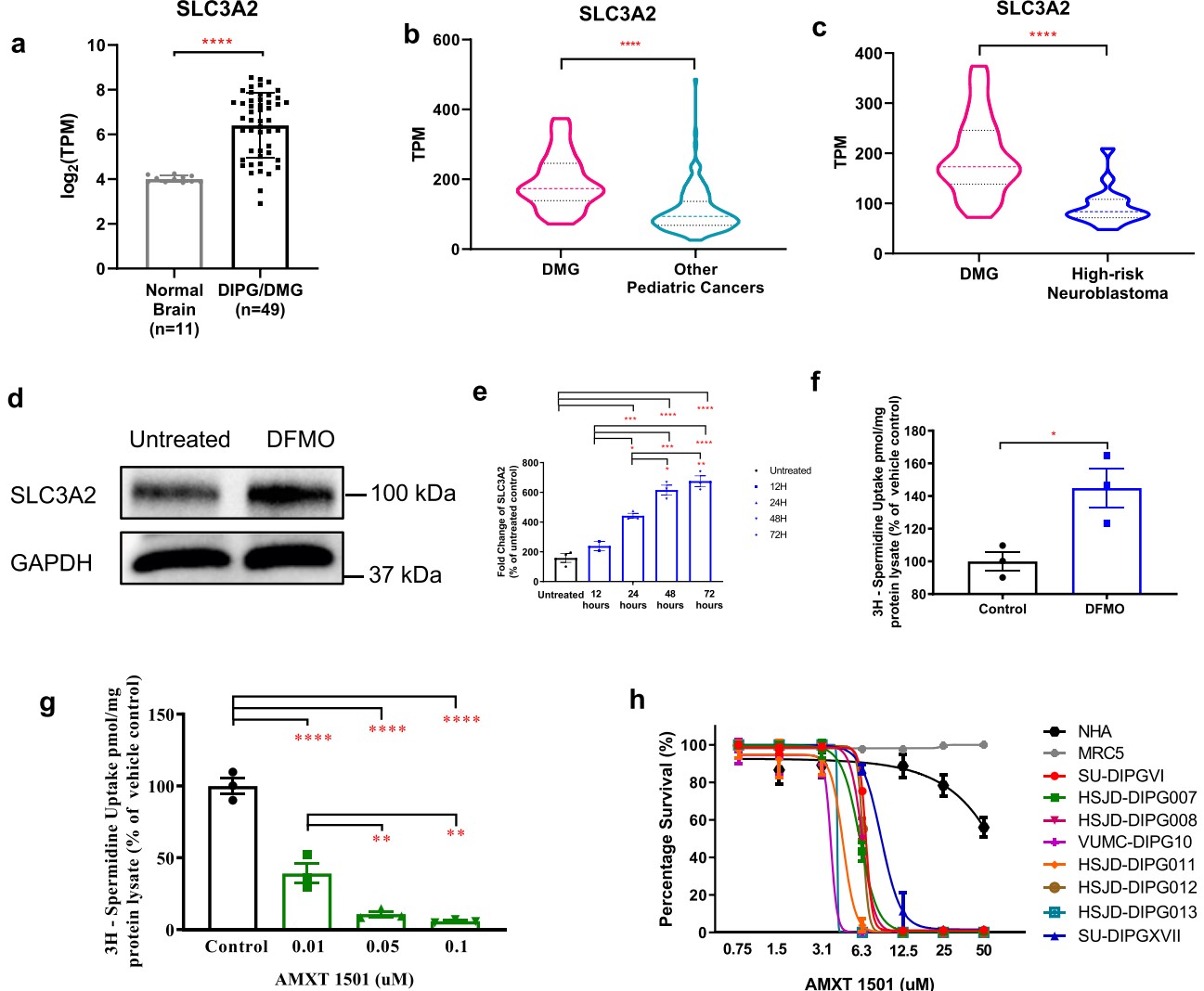

**Fig. 2 Polyamine transporter expression in pediatric DIPG brain tumors.** Expression of polyamine transporter SLC3A2 in (**a**) DIPG tumors ($n = 49$) compared to normal fetal brain ($n = 11$) ($p < 0.0001$). Data were obtained from ZCC/PRISM Clinical trial and McGill University[30]. **b–c** Examination of RNA expression levels in a cohort of high-risk childhood cancers showed that the polyamine transporter, SLC3A2, was significantly overexpressed in DMG ($n = 28$) compared with high-risk childhood cancers ($n = 148$) ($p < 0.0001$) and neuroblastoma ($n = 17$) ($p < 0.0001$). Treatment of HSJD-DIPG007 cells with 40 mM DFMO leads to increased SLC3A2 (**d**) protein and (**e**) gene expression, resulting in (**f**) increased uptake of radiolabeled spermidine ($p = 0.0276$). **g** Polyamine transport inhibition by AMXT 1501 decreased uptake of radiolabeled spermidine and (**h**) was cytotoxic against patient-derived DIPG cell lines. Data are presented as mean values ± SEM of three independent experiments. *$p < 0.05$, **$p < 0.01$, ***$p < 0.001$, ****$p < 0.0001$. **a**, **b**, **c**, **f** Statistical analysis was calculated using two-tailed $t$ tests between sample cohorts. **e**, **g** Statistical analysis was calculated using one-way ANOVA between cohorts and for treated and untreated samples. **e** UT vs 24 h: $p = 0.0005$, UT vs 48 h: $p < 0.0001$, UT vs 72 h: <0.0001, 12 h vs 24 h: $p = 0.0107$, 12 h vs 48 h: $p = 0.0001$, 12 h vs 72 h: $p < 0.0001$, 24 h vs 48 h: $p = 0.0142$, 24 h vs 72 h: $p = 0.0021$. **g** Control vs 0.01: $p < 0.0001$, Control vs 0.05: $p < 0.0001$, Control vs 0.01: $p < 0.0001$, 0.01 vs 0.05: $p = 0.0092$. 0.01 vs 0.1: $p = 0.0026$.

## SU-DIPGVI

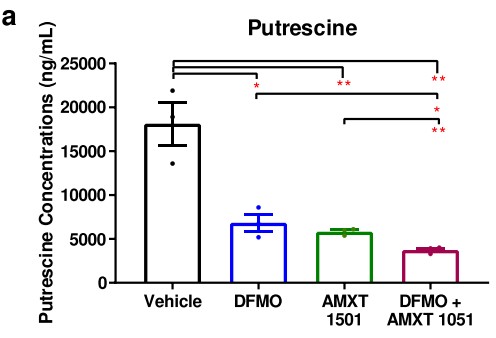

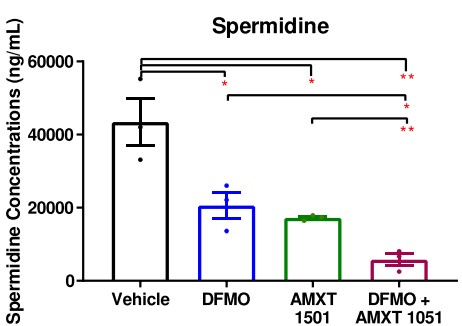

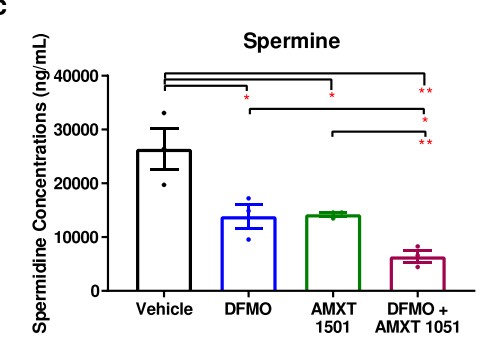

| Polyamine | Synergy Index | |
|---|---|---|
| Putrescine | 1.0 | Additive |
| Spermidine | 0.79 | Synergistic |
| Spermine | 0.82 | Synergistic |

**Fig. 3 Inhibition of synthesis and transport depletes intracellular polyamines. (a)** Putrescine, **(b)** Spermidine, and **(c)** Spermine levels in SU-DIPGVI cells treated with DFMO and AMXT 1501 alone and in combination. Data are presented as means ± SEM of three independent experiments. *$p < 0.05$, **$p < 0.01$. p-values were calculated using one-way ANOVA for untreated and treated samples. Synergy calculated using the Chou–Talalay method. **a** Vehicle vs DFMO: $p = 0.0125$, Vehicle vs AMXT 1501: $p = 0.0072$, Vehicle vs DFMO + AMXT 1501: $p = 0.0041$, DFMO vs DFMO + AMXT 1501: $p = 0.0378$, AMXT 1501 vs DFMO + AMXT 1501: $p = 0.0028$. **b** Vehicle vs DFMO: $p = 0.0364$, Vehicle vs AMXT 1501: $p = 0.0152$, Vehicle vs DFMO + AMXT 1501: $p = 0.0047$, DFMO vs DFMO + AMXT 1501: $p = 0.0212$, AMXT 1501 vs DFMO + AMXT 1501: $p = 0.0027$. **c** Vehicle vs DFMO: $p = 0.0492$, Vehicle vs AMXT 1501: $p = 0.0351$, Vehicle vs DFMO + AMXT 1501: $p = 0.0077$, DFMO vs DFMO + AMXT 1501: $p = 0.0421$, AMXT 1501 vs DFMO + AMXT 1501: $p = 0.0026$.

combined treatment with both AMXT 1501 and DFMO (Supplementary Fig. 11a–c). Thus, these findings suggest that effective polyamine depletion requires targeting of both polyamine synthesis and uptake in DIPG cells.

**Combination of DFMO with AMXT 1501 reduces polyamine synthesis and transport and enhances cell death.** Given these results, we next investigated the effect of dual inhibition of polyamine synthesis and transport inhibition in DIPG cells. The combination treatment of DFMO and AMXT 1501 led to significantly decreased levels of putrescine, spermidine, and spermine compared to either treatment alone (Fig. 3, Supplementary Fig. 12). This resulted in synergistic inhibition of cell proliferation and colony formation across a panel of different DIPG and thalamic DMG neurosphere-forming cultures (AUS-DIPG-017, P001805, Supplementary Table 5), as indicated by combination indexes (CI) of less than 0.6 (Fig. 4a–c; Supplementary Fig. 13a–d). To further confirm that AMXT 1501 is able to sensitize DIPG cells to DFMO treatment we evaluated the response to DFMO alone and combination treatment upon exogenous addition of polyamines. We found that the addition of AMXT 1501 was able to overcome the resistance to DFMO treatment in DIPG cells that were supplemented with polyamines (Supplementary Fig. 14). We did not see any effect on the cell cycle (Supplementary Fig. 15) apart from an increase in the sub G1 population from 24–48 h, and a potent induction of apoptosis 24 h post-treatment by AnnexinV-FITC and propidium iodide staining (Fig. 4d and Supplementary Fig. 16a). This corresponded with elevated cleaved-PARP and caspase-8

(Fig. 4e, Supplementary Fig. 16b and Supplementary Fig. 17) in two primary DIPG cultures. Together, these results show that a combined polyamine depleting strategy has a potent effect on DIPG growth, proliferation, apoptosis, and cell survival.

**Dual inhibition of polyamine synthesis and transport enhances survival in orthotopic models of DIPG.** To evaluate the therapeutic efficacy of the polyamine targeting strategy in vivo, we first performed toxicity studies of AMXT 1501. Animals were treated with a variety of AMXT 1501 doses (5, 7.5, and 10 mg/kg/day) with no change in clinical parameters, and biochemical analysis showed minimal changes with exception of reduced glucose levels at all AMXT 1501 concentrations and lower alkaline phosphatase levels at the highest AMXT 1501 concentrations (Supplementary Table 6). Animals treated with a combination of DFMO and AMXT 1501 (5 and 7.5 mg/kg/day) showed no change in biochemical markers apart from lower glucose levels (Supplementary Table 6). As such a lower dose of AMXT 1501 of 2.5 mg/kg/day, already established to be well tolerated, was used for treatment studies. To assess efficacy we used 3 molecularly distinct DIPG patient-derived cells grown as orthotopic xenograft models SU-DIPGVI-LUC, HSJD-DIPG007, and RA055 (Supplementary Table 5). These models recapitulate the diffuse infiltration seen histologically in DIPG tumors[34,35]. One of the reasons that treatments for DIPG have failed in the clinic is thought to be due to their failure to penetrate the BBB[36]. To confirm the integrity of the blood–brain barrier in these models, we measured the extravasation of Evans Blue (EB) following intravenous

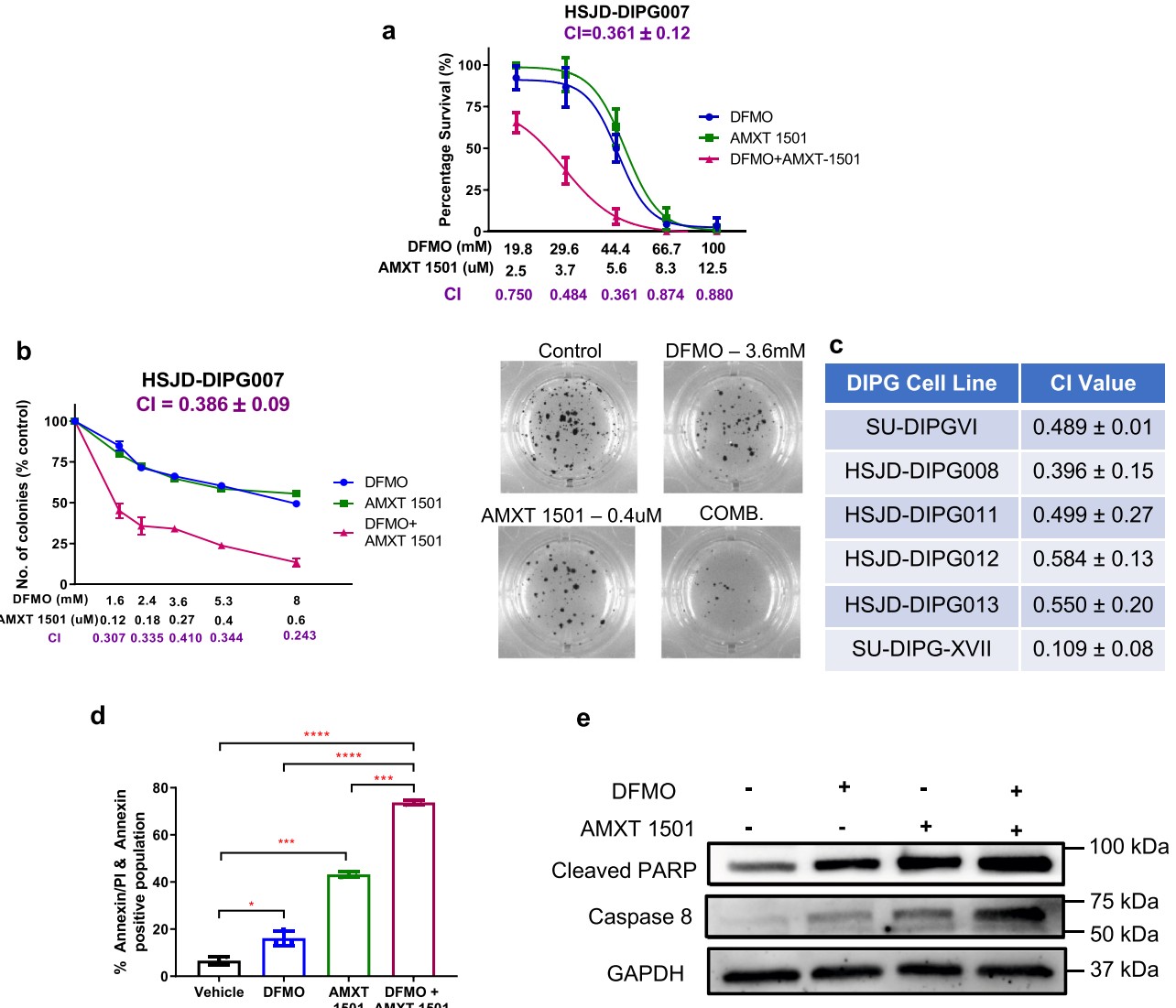

**Fig. 4 The effect of combined inhibition of polyamine synthesis and uptake, using DFMO and AMXT 1501, on cell proliferation, colony formation and apoptosis.** HSJD-DIPG007 cells have synergistically decreased (**a**) cell survival and (**b**) form fewer colonies, upon combination treatment. **c** Synergy scores, calculated by Calcusyn, across a panel of neurosphere-forming DIPG cells treated with combination of both agents. **d** Annexin/PI staining of HSJD-DIPG007 cells treated with 40 mM DFMO, 5 μM AMXT 1501 or combination of both agents for 24 h. Vehicle vs DFMO: $p = 0.0221$, Vehicle vs AMXT 1501: $p = 0.0003$, Vehicle vs DFMO + AMXT 1501: $p < 0.0001$, DFMO vs DFMO + AMXT 1501: $p < 0.0001$, AMXT 1501 vs DFMO + AMXT 1501: $p = 0.0005$. **e** Apoptotic effects of DFMO and AMXT 1501 combination on protein expression of cleaved PARP and caspase 8. Representative blot from two independent experiments. Data are presented as mean values ± SEM of three independent experiments. **a**, **b** $n = 3$ wells examined over three independent experiments. **d** $n = 3$ independent experiments. $**p < 0.01$, $***p < 0.001$. $p$-values were calculated using two-tailed $t$ tests for treated and untreated cohorts.

administration. While there was profound extravasation of EB dye into all organs and skin (Fig. 5a, b), no significant change was seen in the brainstem and cortical regions (Fig. 5b, c). Furthermore, we observed no difference between uninjected animals, matrigel injected and DIPG injected in the brainstem indicating no leakiness in BBB as a result of intracranial injections or DIPG tumor growth (Fig. 5c). Analysis of brain samples from Balb/C nude mice xenografted with the aggressive SU-DIPGVI cells after one week of treatment revealed good penetration of both drugs in the brainstem region at a dose of 2.5 mg/kg (Supplementary Fig. 18). In the same SU-DIPGVI-Luciferase model, DFMO and AMXT 1501 given as a monotherapy each had limited or no antitumor effect, respectively, while the combination significantly extended survival, with 6 out of 9 mice surviving untill the endpoint of 160 days ($p < 0.0001$; Fig. 6a, Supplementary Table 7). Xenogen monitoring of the tumors throughout the treatment period revealed a near complete cessation of tumor growth compared with vehicles (Fig. 6b, c). This result was replicated in second patient-derived DIPG cells grown as an orthotopic animal model (HSJD-DIPG007) where two thirds of the mice survived until the endpoint (Fig. 6d, Supplementary Table 8). Immunohistochemistry of brain tissue collected after 4 weeks of combination treatment revealed significantly decreased cell proliferation as measured by Ki67 staining (Fig. 6e) while western blot analysis of extracted tumor showed increased levels of caspase 8 (Supplementary Fig. 19). Given that radiotherapy is the only current treatment for DIPG, we evaluated whether the combination of DFMO/AMXT 1501 could be further enhanced upon the addition of irradiation. Using the biopsy-derived primary DIPG culture RA055 (Supplementary Table 5) we confirmed using soft agar clonogenic assays that both polyamine pathway inhibitors together with irradiation reduced the clonogenic potential of

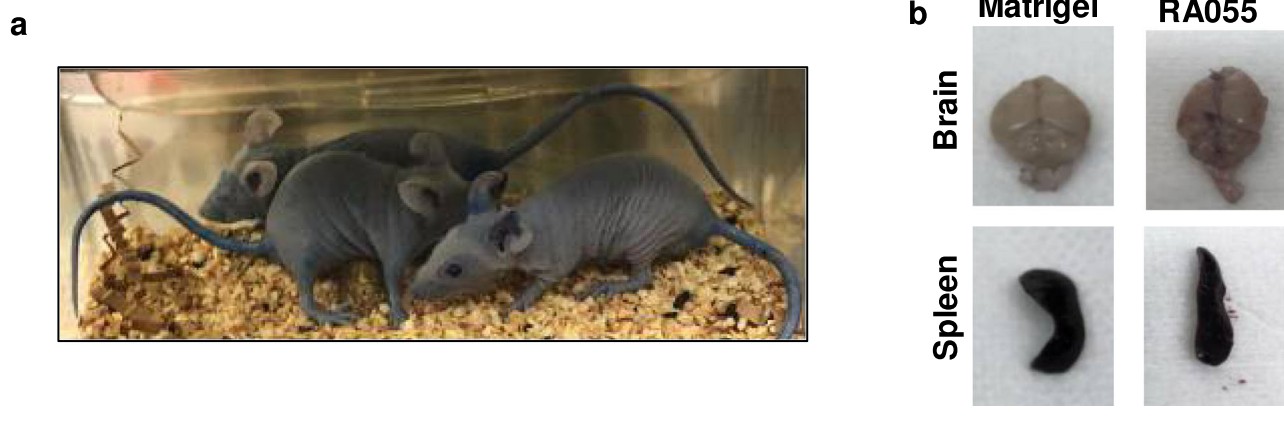

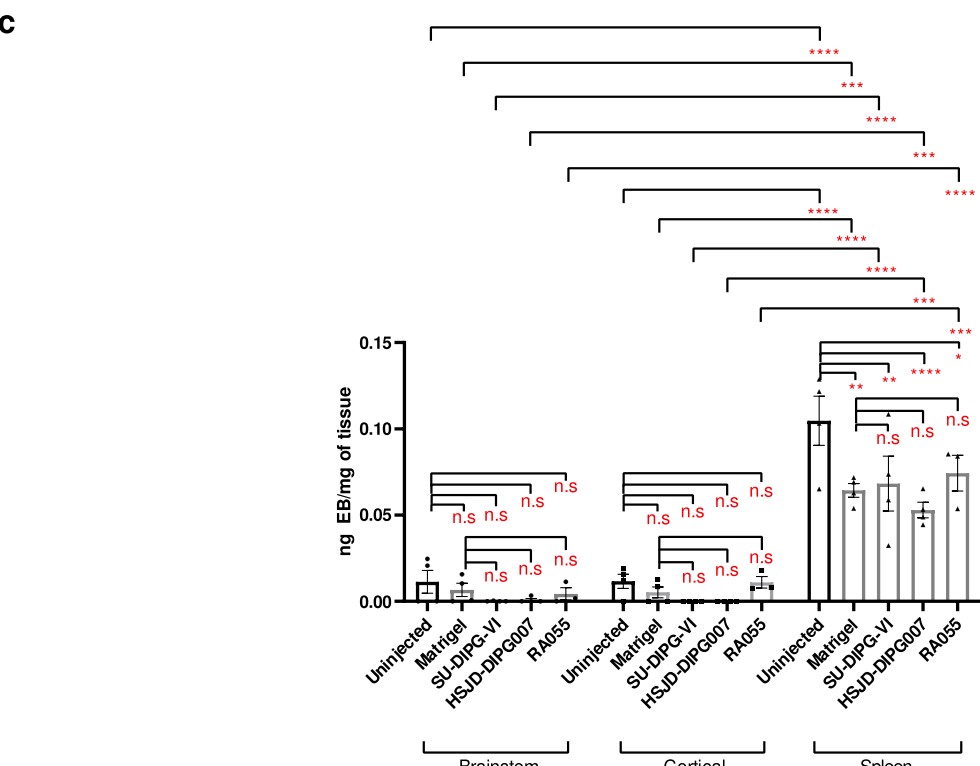

**Fig. 5 Evans Blue (EB) extravasation assay in three orthotopic models of DIPG. a** Representative images of Balbc/Nude mice injected with Evans Blue dye. **b** Representative images of brains and spleens harvested from animals intracranially injected with matrigel and RA055 cells. **c** Brainstem region shows low EB extravasation with no difference among the uninjected, matrigel injected and DIPG injected animals (RA055, SU-DIPGVI, HSJD-DIPG007) and no differences were observed with the tumor free cortical region; Splenic tissue shows higher EB extravasation compared to both brain regions. Data are presented as mean values ± SEM of samples collected from three mice in each cohort. $*p < 0.05$, $**p < 0.01$, $***p < 0.001$, $****p < 0.0001$. Statistical analysis was performed using two-way ANOVA. Spleen-uninjected vs matrigel: $p = 0.0012$, uninjected vs SU-DIPGVI: $p = 0.0040$, uninjected vs HSJD-DIPG007: $p < 0.0001$, uninjected vs RA055: $p = 0.04114$. Brainstem vs Spleen-uninjected: $p < 0.0001$, matrigel: $p = 0.0001$, SU-DIPGVI: $p < 0.0001$, HSJD-DIPG007: $p = 0.0005$, RA055: $p < 0.0001$. Cortical vs Spleen-uninjected: $p < 0.0001$, matrigel: $p < 0.0001$, SU-DIPGVI: $p < 0.0001$, HSJD-DIPG007: $p = 0.0003$, RA055: $p = 0.0002$.

DIPG cells (Supplementary Fig. 20a, b). Furthermore enhanced cleaved parp and phosphorylated H2AX was observed upon the combination of irradiation with both polyamine inhibitors in DIPG cells (Supplementary Fig. 20c). To extend these in vitro findings, we evaluated the therapeutic efficacy of the triple combination in vivo using the same RA055 cells in an orthotopic DIPG model. This highly aggressive model has a median survival of 44 days post intracranial injection in untreated mice. DFMO/

AMXT 1501 treated mice showed significantly enhanced survival with no significant toxicity (Fig. 6f). The addition of irradiation enhanced the survival of DIPG xenografts further, and median survival was not reached (Fig. 6f, Supplementary Table 9). Overall we have found the combination of both polyamine inhibitors to be well tolerated in DIPG orthograft models with single agent and combination treatment cohorts showing stable weights compared to vehicle treatments (Supplementary Fig. 21). Each model tested

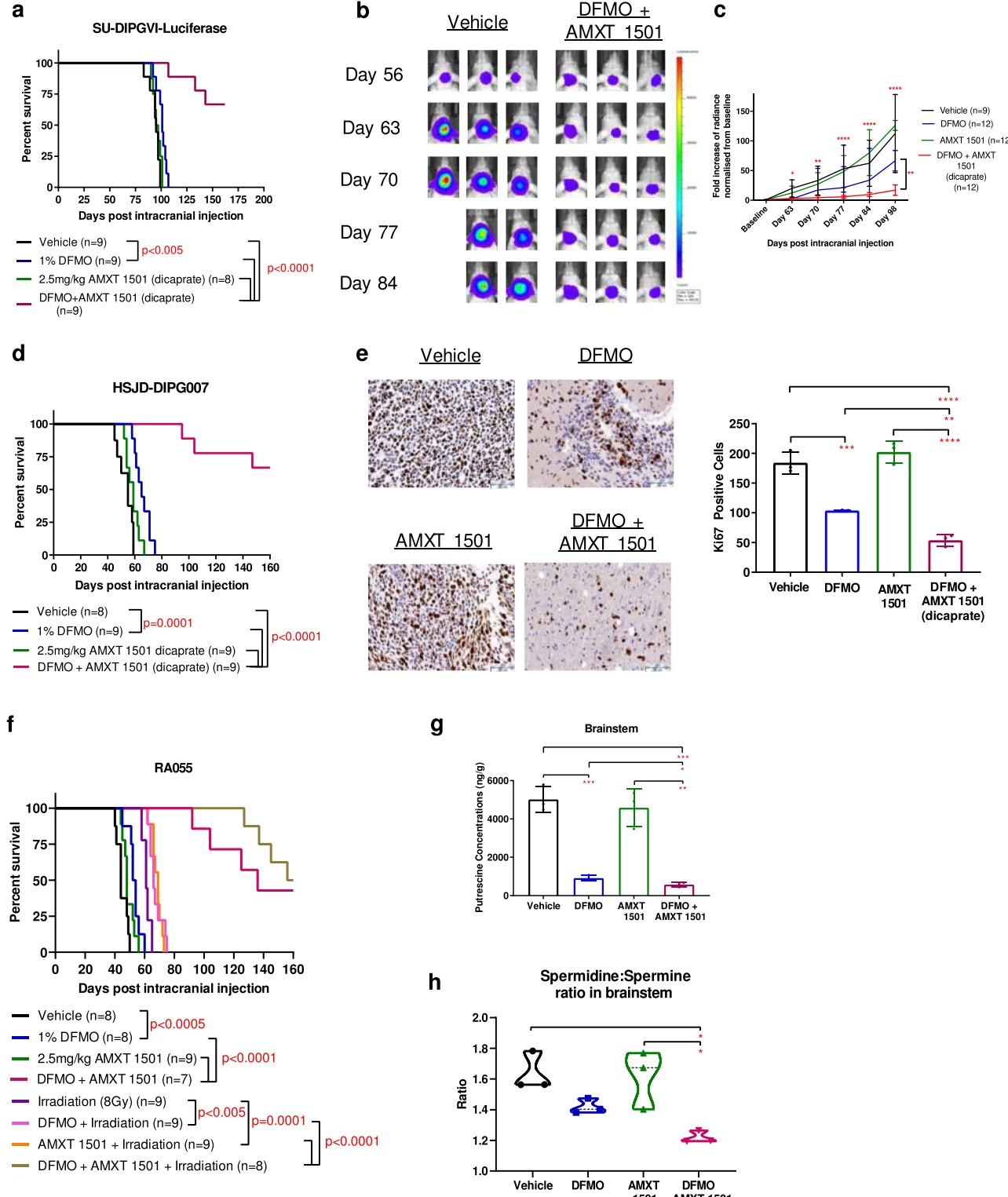

to date has different molecular drivers (Supplementary Table 5), suggesting this treatment may be effective across a broad range of DIPG tumors.

In order to confirm that the therapeutic efficacy of the DFMO/ AMXT 1501 combination is due to pharmacological targeting of the polyamine pathway in the brain, we evaluated polyamine levels in the brainstem of treated mice. While DFMO treatment resulted in decreased putrescine and spermidine levels in the tumor-ridden brainstem area, the DFMO and AMXT 1501

combination led to significant depletion of putrescine and a significantly lower putrescine to spermidine ratio (Supplementary Fig. 22) in comparison with DFMO alone (Fig. 6g). Furthermore a significantly lower spermidine to spermine ratio was observed in combination treated animals in comparison with AMXT 1501 alone (Fig. 6h) indicating reduced polyamine synthesis. In addition we evaluated for effects of polyamine inhibitors on SLC3A2 protein levels in vivo following one week of treatment. As expected we observed significantly higher levels of SLC3A2

**Fig. 6 Therapeutic efficacy of DFMO, AMXT 1501 and combination treatment in orthotopic models of DIPG.** Mice were intracranially injected with DIPG cells and 4–8 weeks post injection treatments commenced. Mice were humanely euthanised when they displayed severe neurological decline and/or weight loss or reached maximum holding time (MHT). **a** Survival curve of SU-DIPGVI-Luciferase with DFMO/AMXT 1501 treatment. Median survival of cohorts (days): Vehicle = 95, DFMO = 101, AMXT 1501 = 96, DFMO + AMXT 1501 = undefined. Exact $p$-values listed in Supplementary Table 7. **b–c** Xenogen imaging showed the cytostatic effect of DFMO/AMXT 1501 on SU-DIPGVI DIPG cells. Vehicle vs DFMO/AMXT1501—Day 63: $p = 0.02301$, Day 76: $p = 0.0024$, Day 77: $p < 0.0001$, Day 81: $p < 0.0001$, Day 98: $p < 0.0001$. DFMO vs DFMO/AMXT1501—Day 98: 0.0050. **d** Survival curve of HSJD-DIPG007 with DFMO/AMXT 1501 treatment. Median survival of cohorts (days): Vehicle = 55, DFMO = 65, AMXT 1501 = 59, DFMO + AMXT 1501 = undefined. Exact $p$-values listed in Supplementary Table 8. **e** Ki67 staining of DIPG tumors post 4 weeks of treatment in HSJD-DIPG007 model. Three images were taken from brain samples collected from three mice in each cohort. Scale bars: black 50um. Vehicle vs DFMO: $p = 0.0005$, Vehicle vs DFMO/AMXT1501: $p < 0.0001$, DFMO vs DFMO/AMXT1501: $p = 0.0099$, AMXT1501 vs DFMO/AMXT1501: $p < 0.0001$. **f** Survival curve of RA055-xenografted mice with DFMO/AMXT 1501 treatment. Median survival of cohorts (days): Vehicle = 44, DFMO = 53, AMXT 1501 = 48, Irradiation = 61, DFMO + Irradiation = 66 days, AMXT 1501/Irradiation=69 days, DFMO + AMXT 1501 = 136, DFMO + AMXT 1501+Irradiation = 158. Exact $p$-values listed in Supplementary Table 9. **g** Putrescine levels in the tumor-ridden brainstem region in SU-DIPGVI-LUC model, post 1 week treatment. Brain samples were collected from three mice in each cohort. Vehicle vs DFMO: $p = 0.0005$, Vehicle vs DFMO/AMXT1501: 0.0004, DFMO vs DFMO/AMXT1501: $p = 0.0315$, AMXT1501 vs DFMO/AMXT1501: $p = 0.0021$. **h** Spermidine to spermine ratio (spd:spm). Brain samples were collected from three mice in each cohort. Vehicle vs DFMO/AMXT1501: $p = 0.0110$, AMXT1501 vs DFMO/AMXT1501: $p = 0.0146$. **a, d, f** Shaded areas indicate treatment period of combination treatment, which was continuous in RA055 model. Statistical analysis has been performed using the Log-Rank (Mantel–Cox) with multiple test correction applied. **c, e, g, h** Data are presented as mean values ± SEM. *$p < 0.05$, **$p < 0.01$, ***$p < 0.001$. $p$-values were calculated using two-way ANOVA (**c**) and one-way ANOVA (**e, g, h**) for treated and untreated cohorts.

upon treatment with DFMO and a subsequent reduction upon combination treatment (Supplementary Fig. 23). Urinary and plasma polyamine concentrations have been shown to be elevated in cancer patients and to correlate with poor prognosis[37–40], while treatment with DFMO led to decreased polyamine levels[41]. We also found that urinary putrescine and spermine concentrations were significantly increased in mice with orthotopic DIPG compared to mice without tumors ($p < 0.05$), and conversely decreased with DFMO and combination treatment (Supplementary Fig. 24). A similar trend was observed in the polyamine levels of plasma (Supplementary Fig. 25). These results raise the potential for urinary and/or plasma polyamines to be used as pharmacodynamic biomarkers for polyamine depleting therapies in DIPG.

Protein translation has been shown to be regulated by polyamines through two separate pathways involving the mammalian target of rapamycin complex (mTORC1) and hypusination of eukaryotic initiation factor 5A (eIF5A)[42]. We firstly dermined for effects on Lin28B/let7 axis, a process directly regulated by hypusinated eIF5A. We found reduced mRNA levels of Lin28b in DFMO and combination treated samples whereas SLC3A2 inhibition increased Lin28B expression levels (Supplementary Fig. 26a). Consistent with the repressive role of Lin28B we observed an increase of Let7 miRNA levels in DFMO and combination treated animals whereas AMXT 1501 treatment led to a reduction in Let7 miRNA (Supplementary Fig. 26a). An alternative pathway which can influence protein translation is mTORC1. We observed decreased phosphorylation levels of mTOR and its downstream target eukaryotic translation initiation factor 4E binding protein 1 (4EBP1) (Supplementary Fig. 26b). These results indicate that polyamine depletion therapy can influence protein translation levels which may impact on DIPG cell proliferation. MYC levels have been associated with the regulation of polyamine pathway and subsequently protein translation[42]. We determined the mRNA copy levels of MYC and MYCN in SU-DIPGVI and HSJD-DIPG007 and found low copy numbers of both genes in comparison to known amplified neuroblastoma cultures (SJ-G2 and BE2C) (Supplementary Fig. 27). Our analysis indicates that the effects we observed in Lin28B/Let7 and mTORC1 pathway upon DFMO and AMXT 1501 treatment are likely independent of the MYC/MYCN status.

## Discussion

To date, there have been over 250 clinical trials for DIPG that have failed, with radiation therapy remaining the only therapy to show clinical activity[43]. In recent years the development of pre-clinical models, including orthotopic animal models, has offered new avenues for discovery[7]. The activity shown here of combined DFMO / AMXT 1501 treatment exceeds that seen in most regimens tested in vivo to date. GD2 targeted CAR T cells showed similar profound efficacy in in vivo DIPG models, but with significant toxicity[12]. Clinically DFMO is well tolerated even at very high intravenous doses, with the most commonly noted side effect being reversible ototoxicity[44]. Other reported side effects such as transaminitis and neutropenia are less common and mainly observed in heavily pretreated neuroblastoma patients[45]. Notably, toxicity was minimal in our animal models with the DFMO/AMXT 1501 combination. Our results suggest that DIPG tumors are critically dependent on elevated polyamine pathway activity, with DIPG cell proliferation driven by the addition of polyamines and potently inhibited by dual targeting of the polyamine pathway.

The over-expression of the SLC3A2 polyamine transporter is likely an important factor leading to the sensitivity of DIPG to this therapy. Remarkably, SLC3A2 levels are increased in both DIPG and other high-grade gliomas (HGG) with significantly higher levels than in other high-risk childhood cancers. The activity of the DFMO/AMXT 1501 combination in DIPG exceeds that previously observed in neuroblastoma models[29]. SLC3A2 is significantly overexpressed in DIPG and HGG compared with high-risk neuroblastoma, which may explain the enhanced in vivo efficacy seen in DIPG. In neuroblastoma, MYCN was recently found to regulate the entire polyamine metabolic pathway[29]. MYC is an uncommon driver of DIPG, and of the DIPG models tested here, SU-DIPGVI has a single copy of MYCN and c-MYC, while HSJD-DIPG007 is c-MYC amplified with a single MYCN copy, suggesting other driving factors in this tumor. Notably, SLC3A2 is significantly overexpressed in both DIPG and HGG compared with other brain and pediatric cancers. Given the heterogeneity of DIPG/HGG tumors[46] and the fact that SLC3A2 over-expression is independent of K27M status, is suggesting that this polyamine depleting strategy could potentially be applicable across a broad spectrum of pediatric HGG including thalamic gliomas although further studies are needed to confirm sensitivity to polyamine depletion therapy.

The in vivo efficacy demonstrated in this study using three aggressive orhtotopic models of DIPG suggests that the combination of DFMO and AMXT 1501 is one of the most efficacious drug combinations tested in DIPG models to date[4,47]. However given that DIPG is an extremely aggressive brain tumor, further

combinatorial approaches with chemotherapy or targeted agents may lead to greater benefit for the patients. We have demonstrated enhanced efficacy of polyamine inhibitors with irradiation, the only current standard therapy for DIPG. The combination of DFMO/AMXT 1501 with chemotherapy (topotecan and cyclophosphamide) has been demonstrated in preclinical models of neuroblastoma[29]. The effects seen on the mTOR pathway suggest that further studies are warranted to determine whether combination with targeted agents such as mTOR inhibitors may further enhance efficacy[10,35,47].

Protein translation is one of the mechanisms that has been suggested for polyamines to influence cell proliferation although the exact mechanisms have not been fully established[42]. Particularly the hypusination of eukaryotic translation initiation factor 5A (5IF5A) can influence MYCN trhough the Lin28B/let7 axis. In neuroblastoma, DFMO has been found to reduce Lin28B protein and increase Let-7 miRNA levels[48,49]. In addition to this polyamines may influence oncogenic pathways such as mTORC1 which plays a key role in supporting protein synthesis. Our work has demonstrated a potential concurrence of both pathways as observed with an increase in Lin28B protein levels as well as decreased phosphorylation of mTORC1 and 4EBP1. Further studies are needed to understand how polyamine inhibition influence protein translation machinery and how this may be therapeutically exploited.

Polyamine biosynthesis is interconnected with other metabolic pathways such as the methionine cycle and arginine metabolism. Recent studies have demonstrated in prostate cancer enhanced metabolic flux of acetylated polyamines to the epithelium in addition to the high demand for polyamine synthesis. This suggested a metabolic sensitivity due to heavy reliance on methionine cycle and methionine salvage pathway which recycles one carbon unit obtained during polyamine biosynthesis[50]. Pharmacological inhibition of the methionine salvage pathway and polyamine acetylation led to reduced tumor growth demonstrating the potential of employing therapies that target metabolic vulnerabilities[50]. Furthermore, in mesothelioma the absence of arginine biosynthetic pathway was found to compensate for polyamine levels though elevated polyamine synthesis. The combination of DFMO with arginine catabolising enzyme (ADI-PEG20) was found to be synthetically lethal in mesothelioma cells[51]. Similarly in other tumors such as neuroblastoma and acute lymphoblastic leukemia, arginine depletion therapy with BCT-100 prolonged survival of animals[52,53]. Future studies are required to understand other metabolic vulnerabilities which can be exploited therapeutically in combination with polyamine depletion therapy.

Our study provides compelling evidence that polyamine synthesis is upregulated in DIPG tumors and that polyamine depletion therapy is an effective therapeutic approach leading to significant tumor growth delay in orthotopic animal models of DIPG. The first clinical trial testing the safety and tolerability of an oral formulation of AMXT 1501 alone and in combination with DFMO is underway in adult solid malignancies (NCT03536728). A pediatric friendly oral formulation is currently in development, offering the potential for translation of this polyamine depleting therapy to the clinic for DIPG patients.

## Methods

**Human DIPG neurosphere-forming cultures**. Patient-derived DIPG and thalamic cells were grown in cancer stem cell (CSC) media consisting of a 50:50 mixture of DMEM/F12 and Neurobasal medium (Invitrogen) supplemented with heparin (Stem Cell Technologies), glutamax, pyruvate, non-essential amino acids, Hepes buffer and antibiotic/antimycotic (Invitrogen). In order to grow the primary cells as neurospheres, CSC media was also supplemented with growth factors such as human EGF, human basic FGF, PDGF-AA, and PDGF-BB (Stem Cell Technologies). Human healthy lung fibroblasts (MRC5) (ATCC) and normal healthy astrocytes (NHAs) (Lonza) were cultured according to the manufacturer's

instructions. RA038 and P000302 cultures were developed from ZCC/PRISM clinical trial as described in[7] and grown in CSC media with the addition of 5% fetal calf serum. DIPG and normal cells were cultured in T75 cm$^2$ flasks at 37 °C in a humidified atmosphere with 5% $CO_2$. ZCC/PRISM clinical trial is approved by the Hunter New England Human Research Ethics Committee. Written informed consent was received from all participants. Specific information on tissue type each culture is derived from, in mutational status, as well as patient treatment information is included in Supplementary Table 5.

**Apoptosis flow-cytometric assays**. DIPG cells were plated at the cell density of 250,000 cells per well and cultured for 72 h to form neurospheres. Subsequently, cells were treated with the indicated dose of drugs for 24 h. Following treatment cells were collected, washed once in cold PBS, and resuspended in 100 μL of Annexin-binding buffer. Cells were stained with 5 μl of Annexin V-FITC and 5 μl of 7AAD (BD Biosciences) for 15 min in the dark and subsequently were diluted to a total volume of 500uL in Annexin-binding buffer. Fluorescence-activated cell sorting (FACS) analysis was performed on Facs Canto (BD Bioscience), to a total of 10,000 events for each sample. Gating strategy utilized shown in Supplementary Fig. 36.

**Cell cycle assays**. A total of 250,000 DIPG cells were plated and cultured for 72 h. Subsequently, cells were treated with the indicated dose of drugs and incubated for either 18, 21, 24, or 48 h. Cells were treated with 0.05% trypsin EDTA, then washed with ice-cold PBS twice and incubated in 70% ethanol at 4 °C overnight. The cell pellets were collected and washed twice with cold PBS. The cell pellets were then resuspended and incubated in staining buffer containing RNase A (100 μg/ml; Roche Diagnostics, Castle Hill, NSW) and propidium iodide (25 μg/ml; Sigma Aldrich, Australia) at 37 °C for 30 min. After centrifugation, the staining buffer was discarded and the pellet was resuspended in cold PBS. The cell cycle was analyzed by flow cytometry(BD FACS Calibur) and data analyzed using FlowJo$^{TM}$ Software (FlowJo$^{TM}$, Ashland, Oregon, USA).

**Western Blot**. Whole-cell extracts were obtained using RIPA Buffer according to the manufacturer's protocol, (SigmaAldrich) and protein quantified by BCA Protein Acid Assay Kit (Pierce). Proteins were resolved on 12–20% Tris-HCl SDS-PAGE precast gels (Bio-Rad) and blotted (Bio-Rad) as described by the manufacturer's instructions. Cell lysates were analyzed with the following antibodies ODC1 (1:500, Cat No: ab97395), SLC3A2 (1:1000, Cat No: VPA00372), cleaved – PARP (1:1000, Cat No: 9541S), Caspase 8 (1:1000, Cat No: 4790S), Phospho-Histone H2A.X (1:1000, Cat No: 9718), p-m-TOR (1:1000, Cat No: 2971S), m-TOR (1:1000, Cat No: 2983), p-4EBP1 (1:1000, Cat No: 236B4), 4EBP1 (1:1000, 9644S), Actin (1:1000, Cat No: D6A8), GAPDH (1:5000, Cat No:97166), Anti-Rabbit IgG HRP-linked antibody #7074, Anti-Mouse IgG HRP-linked antibody #7076. Original scans for all western blots are provided in Supplementary Figs. 28–35.

**Gene expression analysis**. Real-time PCR was performed to determine the expression levels of ODC1, SAT1, SLC3A2, and LIN28B in DIPG and normal cells. In brief mRNA transcripts were isolated from cells using the Qiagen RNA extraction kit and subsequently reverse transcribed with SuperScript III (Invitrogen). Real time PCR was performed using the Prism 7900 Sequence Detection System (Applied Biosystems). Four housekeeping genes were used to standardize gene expression. Relative mRNA levels were calculated by the comparative threshold cycle method. Each gene is expressed as as fold changes of cycle threshold (Ct) value relative to controls. ODC1, SAT1, SLC3A2, and LIN28B KiCqStart SYBR Green predesigned primers (KSPQ12012) were purchased from Sigma Aldrich. Let-7 miRCURY LNA miRNA PCR primes were purchased from GeneGlobe Qiagen (Product No: 339320). Patient expression data for SLC3A2 was obtained from the ZERO childhood cancer program that has RNA-seq samples on 281 patients (28 DIPG, 148 other pediatric cancers, 17 high-risk neuroblastoma) covering all the major high-risk pediatric malignancies[54]. Patient expression data were also obtained from McGill University (21 DIPG, 11 fetal normal brain)[30]. Expression values are listed in Supplementary Tables 1-4.

**Radiolabelled spermidine transport assays**. Polyamine transport in tumor cells was evaluated as described in published protocols[55,56]. Briefly, cells were plated in triplicate and grown to approximately 70% confluence. After washing with PBS,$^3$H-spermidine (NET-522001MC, spermidine trihydrochloride, [terminal methylenes-$^3$H(N)], specific activity 16.6 Ci/mmol; Perkin Elmer, Boston, MA) was added at 1.0 μM and incubated for 60 min at 37 °C. The cells were washed with cold PBS containing 50 μM cold spermidine once followed by two washes with cold PBS, and lysed in 0.1% SDS/PBS for 10 min. The $^3$H radioactivity in each cell lysate was measured by scintillation counting and normalized to protein.

**Proliferation assays**. The cytotoxic effects of DFMO, AMXT 1501 and combination of both agents was assessed in vitro in DIPG and normal cells using the resasurin assay. In brief, DIPG and normal cells were plated in 96 well plates and allowed to form either neurospheres or adhere at the bottom of the well respectively. Three days post-plating cells were treated with DFMO, AMXT 1501 and in combination over a range of concentrations for 72 h. Cell proliferation was

subsequently assessed with the resazaurin assay (Sigma-Aldrich). Data are presented as percentage viability compared to control. To determine synergy, we employed the median effect method using CalcuSyn software (Biosoft). Combination indices (CI) were calculated for each drug combination, where synergy is indicated by a CI < 1, additive where CI = 1, and antagonism where CI > 1.

**Scratch assay.** Silicone culture inserts (IBIDI, DKSH, #80209) were placed in 12-well plates. DIPG cells were plated using the same media listed above in the presence of 5% fetal calf serum at the cell density of $1 \times 10^5$ on either side of the insert. Following and incubation for 24 h, inserts were removed with forceps. Polyamines or drugs were added into 500 μL media and added into the wells. Photographs of the gap wound were taken at times; 0, 6, and 24 h. Images were analyzed using ImageJ Software (ImageJ, Marlyand, USA).

**Clonogenic assays.** The effects of DFMO and AMXT 1501 on the colony-forming ability of the DIPG cells were assessed by soft agar colony formation assay. The assay was performed in 24-well plates. In each well, 300 μL of 0.6% agar (in culture medium) was layered on the bottom followed by 300 μl of 0.3% agar as the top layer. Approximately 3000 HSJD-DIPG007 or SU-DIPGVI cells were plated with the top layer and treated with indicated doses. Cells were maintained at 37 °C in a humidified 5% $CO_2$ atmosphere for 2–3 weeks. The colonies were counted using MTT and presented as percentage colony formation compared to untreated.

**Orthotopic model/xenogen imaging.** Pathogen-free, 5–7 week-old female Balb/C nude mice were purchased from Animal Resources Center (Perth, Australia) and kept at an ambient temperature of 18–22 °C with a humidity of 45–65% under a 12-hour light cycle (7:00 am–7:00 pm). DIPG cells (200,000 cells) were resuspended in 2 μl of matrigel and injected intracranially into the brainstem of Balb/C nude mice by using a stereotactic device (Kopf Instruments) (coordinates: 0.5 mm lateral to midline, 6mmm posterior to bregma suture and 3.5 mm deep). Treatments commenced at day 56 post intracranial injection for SU-DIPGVI-LUC, at day 30 for HSJD-DIPG007 and at day 14 for the RA055 model. Mice exhibiting clinical signs of neurologic decline such as ataxia, circling, head tilting with or without 20% weight loss were humanely euthanised for histological analysis of tumors. Brains were fixed in 10% formalin neutral buffered solution (Sigma-Aldrich), embedded in paraffin wax and 5-mm sections were cut and mounted on glass slides. Following dehydration, sections were stained with hematoxylin/eosin and Ki67 for histologic examination. 1% DFMO was administered continuously in the drinking water and AMXT 1501 dicaprate was dissolved in 3.3% mannitol and administered subcutaneously at 2.5 mg/kg/day for 4 weeks in the SU-DIPGVI-LUC model and HSJD-DIPG007 models, and continuously in the RA055 orthotopic model. Vehicle mice received normal drinking water, and 3.3% mannitol subcutaneously for 4 weeks in the SU-DIPGVI-LUC and HSJD-DIPG007 models, and continuously in the RA055 orthotopic model. All animal experiments were performed according to the Australian Code of Practice for the Care and Use of Animals for Scientific Purposes under the Animal Research Regulation of the New South Wales (Australia) and under a protocol approved by the Animal Use and Care Committees of University of New South Wales.

**MTD studies.** 5–7 week-old, healthy female BALB/c mice were given the following daily treatments for four weeks: AMXT 1501 at 5 mg/kg, AMXT 1501 7.5 mg/kg, AMXT 1501 10 mg/kg, 1% DFMO + 5 mg/kg AMXT 1501 and 1% DFMO + 7.5 mg/kg AMXT 1501, with three mice in each cohort. The vehicle group received daily subcutaneous injections of mannitol for four consecutive weeks. Following the end of treatment, blood was collected via tail bleeds into heparinized tubes for biochemistry analysis. Blood was obtained from all three mice in each cohort and collated into one sample per each treatment. Blood biochemical markers were measured using the VetScan VS2 (Abaxis, Union City, CA, USA) Comprehensive Diagnostic Profile.

**Evans Blue extravasation assay.** Animals were injected orthotopically with SU-DIPGVI, HSJD-DIPG007, and RA055 primary patient-derived DIPG cells as mentioned above. At 2 weeks (RA055) and 4 weeks (SU-DIPG-VI and HSJD-DIPG007) past intracranial injections, animals were injected with Evans Blue (0.5% w/v filter sterilized). Following 20 min, animals were anesthetized and cardiac perfusions were performed. Evans blue dye levels were measured from the brain regions, and spleen after extraction from the mouse. Brain region samples were manually homogenized in 500 μl 50% (wt/vol) trichloroacetic acid (TCA) in saline and 1 ml 50% TCA for spleen and liver samples. Sample homogenates were centrifuged at 6000 g for 20 min at 4 °C. Supernatants were transferred to new sample tubes and volumes were determined. 100ul aliquots of each sample were transferred to a 96-well plate. Standards were prepared in 50% TCA between concentrations 0 ng/ml to 300 ng/ml. Absorbance was measured at 610 nm using Perkin Elmer Victor3 Multilabel Plate Reader[57,58].

**Drug level determination.** DFMO and AMXT 1501 levels were measured following sample extraction. 1% acidified plasma was prepared from formic acid and sodium citrate plasma. Briefly, samples, on ice, were exposed to solutions of calibrated internal standards (200 ng/mL DFMO-d3 and 7.5 ng/mL AMXT 1501-

13C4) were added to samples following vortexing and centrifugation. Aliquots were then treated to ammonium acetate for DFMO, and formic acid for AMXT 1501, prior to mixing. Samples were analyzed by $LC/MS^2$ by the Discovery Facility at Charles River Laboratory.

**Polyamine Level Determination.** Polyamines were measured following pre-column derivization by borate buffer and benzoyl chloride using stable-labeled polyamine standards. Briefly, samples were homogenized in 1xPBS with 1% folic acid, followed by a 100x dilution in the surrogate matrix. Solutions of calibrated standards (500 ng/mL putrescine, spermidine and spermine) were added to samples followed by mixing and centrifugation. Aliquots were treated with borate buffer and 2% benzoyl chloride in acetonitrile preceding vortexing and centrifugation. Derivatized samples were analyzed by $LC/MS^2$ by the Discovery Facility at Charles River Laboratory.

**Statistical analysis.** All in vitro experiments were performed at least in triplicate, and data were compiled from 2–3 separate experiments. Data were analyzed with GraphPad Prism 5 (Statistical Software for Sciences) using one-way or two-way ANOVA unless otherwise indicated. P values less than 0.05 were considered statistically significant. CI values to determine synergy were determined using Calcusyn software. In vivo studies were carried out using multiple animals ($n = 8$–9 per treatment group). The in vivo efficacy of polyamine inhibitors in the three DIPG xenograft models was assessed using the Kaplan-Meier method. The program GraphPad Prism 5 was used for in vivo statistical analyses using the Mantel-Cox test and multiple test corrections using the two-stage step-up method of Benjamini, Krieger, and Yekutieli[59]. In all cases, values of $p \le 0.05$ (Mantel–Cox) were regarded as being statistically significant. In the later method statistically significant results are flagged as "discoveries".

**Reporting summary.** Further information on research design is available in the Nature Research Reporting Summary linked to this article.

## Data availability

The ZCC RNA-seq data are available in the European Genome-phenome Archive (EGA) database under the accession code EGAS00001004905. These data are available under controlled access. In order to gain access to the data, a request must be made to the Data Access Committee at EGA who will forward the appropriate approval documents that will be reviewed by the ZERO childhood cancer Research Management Committee. All the other data supporting the findings of this study are available within the article and its supplementary information files and from the corresponding author upon reasonable request. A reporting summary for this article is available as a Supplementary Information file.

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

## Acknowledgements

The authors thank A/Prof Monje, Dr Angel Montero Carcaboso and Dr Esther Hulleman for generously supplying the SU-DIPG, HSJD-DIPG, and VUMC-DIPG010 cells respectively The Scientific Services Group, Flow Cytometry/BRIL Facility at UNSW, Dr Jayne Murray and Kathleen Kimpton at CCI for their technical support and advice. This work was supported by grants from the National Health and Medical Research Council, Cancer Institute NSW, the DIPG Collaborative, the Cure Starts Now, Tour de Cure Foundation, Cure Brain Cancer Foundation, Levi's project, Benny Wills Brain Tumor Research fund, and Isabella and Marcus Foundation's funding and Gemma Howell Scholarship.

## Author contributions

M.H., M.D.N., D.S.Z., and M.T. conceived the project. D.S.Z. and M.T. supervised the project. M.R.B. and L.G. advised and provided materials for the project. D.M.T.Y. performed radiolabelled spermidine experiments. A.K. performed the rest of the in vitro work. A.K., M.T., C.U., and D.H.U. performed in vivo experiments. C.M., S.H., N.J., and C.L.K provided gene expression data for the project. R.P. and A.E. provided important insights on experiments. A.K. made the figures. A.K., M.T., and D.S.Z. wrote the manuscript. All authors reviewed the manuscript.

## Competing interests

M.B. is employed as President and CSO of Aminex Therapeutics, Inc. and also has ownership interest, including patents, in the same. All other authors declare no competing interests.
