## [Peer Review File · Nature Communications]

Reviewers' comments:

Reviewer #1 (Remarks to the Author):

Overall a well written paper with impressive survival curves.

Major suggestions:

Since the authors indicate that radiation is the primary therapy for DIPG, and polyamine inhibition induces apoptosis, they should test if radiation therapy combines with polyamine inhibition to further augment apoptosis. The combination of radiation+DFMO+AMXT1501 will likely be tested in human patients.

Are the cell lines used all derived from autopsies? If so, the authors should test their therapies against SF7761, which is biopsy derived and available for commercial purchase. DIPG cells may significantly change their basic biology after surviving the selection pressure of 50+ Gy of radiation. Upregulating the polyamine pathway may be an escape mechanism whereby cells survive radiation, particularly if it holds that polyamines may regulate LIN28B, as was demonstrated in neuroblastoma.

The paper would be strengthened if the authors demonstrated increased uptake of spermidine in more than just HSJD007 (figure 2D).

Do the authors have an explanation for why AMXT1501 has activity in vitro but fails as a single agent to have any effect in vivo?

How do the authors explain the lack of induction of sub G1 population with combination therapy in supplementary figure 9 compared to robust induction of cleaved PARP in supplementary figure 10? Were the subG1 cells gated out of the analysis?

How important is continuous dosing with DFMO? Pediatric patients with neuroblastoma were dosed twice daily for 2 years. It would be valuable to know if the synergistic killing with DFMO and AMXT1501 would allow for periodic treatment since the twice daily DFMO and AMXT1501 regimen proposed in NCT03536728 might impose a very large medicine burden on small children.

The pharmacodynamic measurement of decreased proliferation as measured by Ki67 shown in Figure 4e would be complemented by immunohistochemistry showing induction of CC3 or CC8 after one week (or some other short timecourse) of therapy. Specifically, does the induction of apoptosis that is seen with combination therapy in vitro also manifest after 5-7 days of treatment in vivo?

While the measurements of polyamines in the brainstem definitively show that DFMO depletes these polyamines, there is only a very small additive effect of AMXT1501 to this depletion (which one might expect) and the mechanism of action is lacking in terms of how this combination is suppressing growth in vivo. Is LIN28B expression suppressed with subsequent upregulation of let7? And if so, is there additive or synergistic suppression of this pathway with DFMO and AMXT1501? Are other targets of DFMO such as protein synthesis suppressed in vivo?

Why would DFMO lead to decreased expression of ODC1 mRNA (supplemental figure 4)? DFMO is thought to be an inhibitor of ODC1 activity and potentially a global translational inhibitor, though it also possibly has effects on nucleotide pools. Is the decrease in ODC1 mRNA a function of decreased cell proliferation and the onset of apoptosis?

The survival curves are impressive – however analysis requires that a multiple comparisons test be applied to the log-rank test. This can be done in GraphPad using “compare a stack of p-values”. After such a multiple comparisons test was undertaken, I highly doubt that DFMO would be found to be statistically significantly extending survival compared to control. The discontinuous X axis employed in figure 4a should not be used, since this is specifically done by the authors to try to highlight the non-clinically significant difference in survival between DFMO and control in SUDIPGVI.

Supplemental figure 6 should be repeated in multiple DIPG cell lines to add robustness to the author’s assertion that DFMO treatment decreases ODC1 protein expression and AMXT1501 increases ODC1 protein expression

Minor suggestions:

MRC5 and their normal human astrocytes (NHA) should be defined better in the methods and described in the figure legends.

The authors make claims about ornithine pathway inhibition in pediatric non-midline GBM – however they do not show data of activity in this manuscript that I can see (there are some supplemental figures showing expression of pathway genes). They should either include data showing similar activity of polyamine pathway inhibition in this paper or should remove those claims.

The authors refer to HSJD-007 as a “PDX” – this implies that it has never been grown in plastic and is serially passaged. I do not believe this is that case and the term “PDX” should be changed to “DIPG patient-derived cell line grown as an orthotopic xenograft”

Please double check the stereotactic coordinates for injection since “coordinates: 0.5 mm anterior, 6.0mm lateral” seems reversed from the usual which would be just lateral to the midline and either 6 mm posterior from bregma or slightly anterior or posterior from lambda.

Reviewer #2 (Remarks to the Author):

Khan and colleagues report on the overactivation of the polyamine synthetic pathway in 35 samples of diffuse intrinsic pontine glioma and show how inhibition of this pathway with DFMO (ODC1 inhibitor) and AMXT 1501 (SLC3A2 transporter inhibitor) results in prolonged survival in cell lines and animal models. The authors show changes in these pathways in some (but not all) cell lines tested. They mention how their treatments had no toxicity on animals, but failed to demonstrate so. Overall, this study raises interesting points on targeting a pathway novel for DIPG (but not novel for other solid tumors), but fails to demonstrate its significance in other midline gliomas and even across all the DIPG cell lines tested. Further, a discussion on the molecular differences existing in DIPG and how these could affect treatment is needed.

General comments:

1. The article would benefit from separation in clear categories (abstract, introduction, results), each with appropriate subheadings. Similarly, the methods should be organized more clearly.
2. The article should be proofread for clarity; punctuation should be corrected.
3. With molecular evidence mounting, DIPG is now a sub-entity of diffuse midline gliomas. This study would benefit from a discussion and demonstration on how its results could be applicable to other midline gliomas (e.g. thalamic).
4. The authors did not discuss the histone 3 status of their cell lines and mouse models. The h3.3k27m mutation has been shown to pertain to worse prognosis in DIPG with treatments against this genetic subtype being developed.
5. Similarly, the authors did not consider DIPG heterogeneity and molecular profiles and how these could affect response to therapy. This paper (10.1016/j.ccell.2017.08.017) should be referenced.
6. The authors briefly touch on BBB permeability. However, they do not demonstrate how the BBB is intact in their animal models nor do they discuss whether DFMO and AMXT 1501 are, in fact, BBB permeable.
7. Most control graphs lack error bars. The authors state that all controls were done in triplicates. However, if that is the case control graphs would still require error bars. We recommend enquiring with a statistician on how to best represent these data (normalize each value against the average of the control, and then normalize, rather than normalizing average against average).
8. The authors mention in passing that their treatment is safe. However, they fail to show so. In clinical trials, DFMO treatment has been shown to have Grade 3 toxicities in children with neuroblastoma, with transaminitis and neutropenia being common side effects. The authors should further test for immune system-related toxicities in immune-competent mice.
9. The toxicity profile of AMXT-1501 is similarly poorly characterized and the field would benefit from an in-depth safety assessment in mice (at least).

Specific comments:

1. Figure 1a and 1b. The authors are unclear on the origin of these samples. Were these DIPG samples obtained from a consortium? It is similarly unclear if 'signal intensity' refers to RNA or protein expression. Further, it needs to be clarified what 'normal brain' refers to: adult brain samples or age-matched pediatric brainstem (how and why was it obtained?)? Further, the use of 35 samples only, in light of papers such as the Mackay discussed above, where ODC1 and SAT1 levels could be obtained (at least on the mRNA level) is rather underwhelming. ODC1 and SAT1 levels should also be compared with other known DIPG mutations and drivers.
2. Figure 1c-1e. The changes in ODC1 and SAT1 are not consistent across cell lines, with some cell

lines not differing from control. Further, using MRC5 as a control is controversial; pediatric brainstem tissue (astrocytes) would be a better control.

3. Figure 2a. Again, the use of 'normal brain' is controversial and should be explained.

4. Figure 2d. The WB is underwhelming and, by eye, hardly significant. A quantification graph is necessary.

5. Figure 2e. The significance of this graph is questionable. The authors show an increase in mRNA expression 12 and 24 hours after treatment. All cell experiments thus far, and all experiments following, have an incubation time of 3 days. This timepoint should be used. Further, the statistics used seem to be incorrect (ANOVA instead of t-test).

6. Figure 2g. The authors used the wrong statistical analysis tool; this graph would benefit from a one-way ANOVA with multiple comparisons to show how increased drug concentrations lead to decreased uptake.

7. Supplementary Figure 7. This figure somewhat undermines the authors' point, as combination treatment is not always superior to each drug alone and, judging from the graphs, not even additive. The authors should repeat these experiments and address their conclusions in light of these findings.

8. Figure 3a. The graph of DFMO is not concordant with the one shown in figure 1h, as the latter graph had an end survival of 20% at the highest concentration. The one in 3a does not.

9. Figure 3a-c. Combination studies are correctly executed across an array of concentrations. CI's, however, are also dependent on, and should be indicated at, different drug concentrations. This is the case because, at extremes of concentration, combinatorial effects might wane. As such, the relevance of the combinatorial indices shown is unclear.

10. Figure 3e. This WB needs quantification as the differences across treatments are not convincing.

11. Page 6. The authors indicate that their mouse models maintain an intact BBB and cite reference 30. Of note, Hennika et al. used a different mouse model from those used in this paper which, nonetheless, had a permeable BBB. Further, it remains the matter of debate whether xenograft mouse models can maintain an intact BBB.

12. Figure 4a, 4d, 4f. These graphs would benefit by indicating the median survival for each treatment group. Further, the authors should discuss the significance of the fact that DFMO treatment alone yield a significant survival benefit compared to control. The timing of treatment is also unclear and should be better explained in the methods.

13. Figure 4b and 4c. These figures lack the single drug control. Further, it would be beneficial to know how the luminescence signal progressed from implantation to when the authors start showing changes.

14. Figure 4g. these comparisons require one-way ANOVA with multiple comparisons rather than repeated t-tests. Further, the authors looked at levels of putrescine alone as a measure of pathway inhibition – the changes in spermidine and spermine are not impressive (if existing). A more in-depth characterization of the in vivo levels of these agents is necessary. Further, it would be beneficial to know the level of ODC1 and SLC3A2 following treatment in vivo.

Reviewer #3 (Remarks to the Author):

In the article by Khan et al, entitled "Dual Targeting of polyamine synthesis and uptake in diffuse intrinsic pontine gliomas", the authors present a combination therapy using DFMO to target

polyamine biosynthesis while simultaneously targeting polyamine import via AMXT 1501. The idea being that cells survive DFMO treatment by compensating through increased uptake of polyamines, thus co-blocking the uptake of polyamines and synthesis will lead to critical polyamine deficit in the cancer cells and be an effective therapy. They demonstrate that such a strategy might be appropriate in DIPG tumors because they have relatively high levels of ODC1 and low levels of SAT1. A panel of patient derived cell cultures showed similar results, grew better with the addition of polyamines to the media, and demonstrated antiproliferative responses to range of doses of DFMO. A key point is that upon DFMO treatment there is an increase in polyamine uptake that could be blocked by use of AMXT 1501. They further demonstrate that combination therapy is more effective than single agent in a series of culture experiments and move to a nice in vivo model. Using multiple PDX models implanted into mouse brainstem they demonstrate that the combination is far more effective than single agent at blocking tumor growth and extending survival.

The major concern with this manuscript is around the question of having sufficient novelty for publication in Nature Communications, as the concept of combining DFMO with AMXT 1501 has been established by many groups. Members of the current author group published this themselves in STM in 2019 for use in neuroblastoma. This manuscript does not appear to add anything to that work other than to be in a different disease site. Nevertheless, there is a paucity of treatment options for DIPGs, and the current study uses nice PDX models to demonstrate efficacy with this combination approach, which is certainly important. Perhaps the concerns about novelty could be allayed by the addition of some mechanistic experiments. Below are some specific concerns:

1) In figure 1f and 4g, the authors measure putrescine levels, but really need to show spm and spd in the main figures. The spd:spm ratio may also be informative. This perhaps would lead to additional experiments that would provide novelty in terms of mechanism. For example, looking at the polyamine levels in Fig 4g and supplemental fig 12a and 12b, there seems to be minimal added change in polyamines when AMXT 1501 is added to DFMO, despite dramatic differences in survival and tumor growth. How do the authors explain this? It may be worthwhile asking what is happening to the PTEN and MTOR pathway, AMD1 activity levels, dcSAM levels, and SMOX and PAOX activity levels? Is there an accumulation of H₂O₂ or indicators of oxidative stress?

2) It is not clear from the methods if there is serum in the media used for studies where polyamines are added to the media to affect growth and migration. If so, is aminoguanidine added to prevent amine oxidase activity.

3) It might be informative to ask if the IC₅₀s defined in Fig 1h correlate with quantification of ODC1 levels by Western in fig 1c.

4) What are the sample sizes in fig 2b and 2c?

5) The Western blot in Fig 2d, meant to demonstrate upregulation of SLC3A2, is not so convincing. Conceptually, it is not important that there is an increase in SLC3A2. It is sufficient to say that it is still there, and therefore targetable. Showing convincingly that it is upregulated after DFMO treatment really does not matter much. One does not need to upregulate the transporter for it to be more used in the context of DFMO treatment. The key point comes in Fig 2f demonstrating

increased uptake, and this is true regardless of whether or not the Western convincingly demonstrates upregulation of the transporter. There is, however, an important missing experiment: If the argument is that DFMO induced increases in uptake help to make cells resistant to DFMO, then if spd/spm are added to the media does this make the cells more resistant to DFMO, and can this be countered by AMXT 1501?

6) Supplemental Fig 7 contains important data that should be included in the main figures for at least one of the models. Going back to point 1, it is interesting that all three polyamines are reduced in these culture models, but it looks very different in vivo. Can the authors explain this?

7) Based on data from Supplemental figure 11, the authors argue that these drugs can cross the blood-brain barrier. It would seem that an important control would be to include animals that have not had anything implanted in their brainstem. This would control for the possibility that the physical process of implantation, or the presence of foreign tumor, influences the ability of the two drugs to cross the blood-brain barrier.

8) Though toxicity of the combination was minimal, how was toxicity measured, what was the toxicity, at what dose, given for how long?

9) The authors state that “Our results suggest that DIPG tumors are critically dependent on elevated polyamine pathway activity, with tumour growth driven by the addition of polyamines...” It was not demonstrated that tumor growth was driven by the addition of polyamines. Fig 1g shows some increased proliferation in culture, but that is very different that the quoted statement.

Title: Dual targeting of polyamine synthesis and uptake in diffuse intrinsic pontine gliomas

We thank the editorial team and the referees for their thoughtful and thorough review and for the helpful and constructive comments. We have now addressed all the comments raised and are pleased to be able to re-submit what we believe is a substantially improved manuscript. We have included new experimental data and incorporated new *in vivo* modelling, experiments showing an intact blood brain barrier, and new toxicity data. The specific changes and responses are addressed below.

Reviewer 1

1. Since the authors indicate that radiation is the primary therapy for DIPG, and polyamine inhibition induces apoptosis, they should test if radiation therapy combines with polyamine inhibition to further augment apoptosis. The combination of radiation+DFMO+AMXT1501 will likely be tested in human patients.

We agree and thank the reviewer for this recommendation. We have now conducted further experiments that confirm the activity of polyamine inhibitors in combination with irradiation. In supplementary Figures 20a,b we show that irradiation synergistically inhibits colony formation in combination with DFMO and AMXT 1501 in HSJD-DIPG007 and RA055 cells *in vitro*. Western blotting analysis demonstrated enhanced cleaved parp and phosphorylated H2AX levels upon combination of irradiation with both polyamine inhibitors (Supplementary Figure 20c). These results are described on page 9, paragraph 1, lines 193-198. In Figure 6f we show the results of an *in vivo* efficacy study using the RA055 animal model. Consistent with the *in vitro* observation, the combination of dual polyamine inhibitors with irradiation (8Gy) significantly enhanced the survival of the animals compared to DFMO/AMXT 1501 treatment. These results are described on page 9, paragraph 1, lines 199-203.

2. Are the cell lines used all derived from autopsies? If so, the authors should test their therapies against SF7761, which is biopsy derived and available for commercial purchase. DIPG cells may significantly change their basic biology after surviving the selection pressure of 50+ Gy of radiation. Upregulating the polyamine pathway may be an escape mechanism whereby cells survive radiation, particularly if it holds that polyamines may regulate LIN28B, as was demonstrated in neuroblastoma.

We agree that it is important to test each therapy in different tumours that may or may not have been treated with radiation therapy. Some of the cultures we have tested were from biopsies, and some from autopsies. We have updated Supplementary Table 1 to include information on the biopsy and autopsy status for all the primary cultures used on this study. The RA055 primary culture (used in both *in vitro* and *in vivo* experiments) is derived from a biopsy from a patient that had not received any prior treatment. The SU-DIPG-VI model is derived from an autopsy of a patient previously treated with radiotherapy and vorinostat (Nagaraja S, et al,

Cancer Cell, 2017). The HSJD-DIPG007 culture is derived from a patient that succumbed to disease within one month from diagnosis and received unspecified treatment (MacKay et al, Cancer Cell, 2017). Together, the *in vitro* and *in vivo* results collectively show that inhibition of the polyamine pathway enhances survival in orthotopic models of DIPG irrespective of tissue type and irradiation status.

3. The paper would be strengthened if the authors demonstrated increased uptake of spermidine in more than just HSJD007 (figure 2D).

We agree with this comment and have now included in Supplementary Figure 10 the effects on polyamine uptake following DFMO and AMXT 1501 treatment of SU-DIPGVI. As observed for HSJD-DIPG007 radiolabelled spermidine uptake is increased following treatment with DFMO whereas uptake is reduced with increasing concentrations of AMXT 1501. These results are described on page 6, paragraph 2, lines 125-127 and 130-132.

4. Do the authors have an explanation for why AMXT1501 has activity in vitro but fails as a single agent to have any effect in vivo?

The precise reason for this difference is not defined, but has also been described in neuroblastoma where, as with DIPG, AMXT 1501 reduced cell proliferation in three NB cell lines (Samal et al International Journal of Cancer, 2013, 133(6): 1323-1333), although no extension in survival was observed in the transgenic animal model TH-MYCN (Gamble, et al Science Translational Medicine, 2019, 11(447)pii).

5. How do the authors explain the lack of induction of sub G1 population with combination therapy in supplementary figure 9 compared to robust induction of cleaved PARP in supplementary figure 10? Were the subG1 cells gated out of the analysis?

The difference in results seen in the cell cycle versus western blot experiments relates to the timing of the assays. We have now repeated cell cycle arrest experiments at 24h and 48h following dual polyamine combination treatment. Although we do not see any effect at early timepoints, we observed a moderate effect at 24h and a significant increase in the Sub G1 population at 48h. These results are now included in Supplementary Figure 15. These results are included in page 7, paragraph 1, lines 154-156.

6. How important is continuous dosing with DFMO? Pediatric patients with neuroblastoma were dosed twice daily for 2 years. It would be valuable to know if the synergistic killing with DFMO and AMXT1501 would allow for periodic treatment since the twice daily DFMO and AMXT1501 regimen proposed in NCT03536728 might impose a very large medicine burden on small children.

Given its short half-life, it is likely that continuous dosing of DFMO, in combination with AMXT 1501 would lead to the greatest efficacy. There is extensive experience with continuous dosing of DFMO in childhood cancer patients. High dose DFMO has been used in relapsed neuroblastoma patients, combined with chemotherapy with children taking oral DFMO three times per day, every day for up to 12 months (Marachilian et al, JCO 2018 36:15_suppl, 10558-10558). The same regimen is now being explored in a randomised Phase 2 trial through the Children's Oncology Group for children with high risk neuroblastoma following their first relapse

(ANBL1831). In this trial children receive continuous oral DFMO taken three times per day for up to 12 months. In other childhood cancers, such as acute lymphoblastic leukaemia, children receive daily oral maintenance therapy for up to 3 years. Given that DIPG is an incurable disease, with no effective treatment options, it is likely that an oral therapy taken two or three times per day would be acceptable to both parents and children.

7. The pharmacodynamic measurement of decreased proliferation as measured by Ki67 shown in Figure 4e would be complemented by immunohistochemistry showing induction of CC3 or CC8 after one week (or some other short time course) of therapy. Specifically, does the induction of apoptosis that is seen with combination therapy in vitro also manifest after 5-7 days of treatment in vivo?

We agree and have now assessed induction of caspase 8 *in vivo* in SU-DIPGVI-engrafted animals while receiving treatment as suggested. We observed increased protein levels of caspase 8 following treatment with DFMO and AMXT 1501, indicating that apoptosis is induced *in vivo* as observed *in vitro*. These results are included in Supplementary Figure 19 and in results section (page 8, paragraph 1, lines 189-190).

8 and 9. While the measurements of polyamines in the brainstem definitely show that DFMO depletes these polyamines, there is only a very small additive effect of AMXT1501 to this depletion (which one might expect) and the mechanism of action is lacking in terms of how this combination is suppressing growth in vivo. Is LIN28B expression suppressed with subsequent upregulation of let7? And if so, is there additive or synergistic suppression of this pathway with DFMO and AMXT1501? Are other targets of DFMO such as protein synthesis suppressed in vivo?

Why would DFMO lead to decreased expression of ODC1 mRNA (supplemental figure 4)? DFMO is thought to be an inhibitor of ODC1 activity and potentially a global translational inhibitor, though it also possibly has effects on nucleotide pools. Is the decrease in ODC1 mRNA a function of decreased cell proliferation and the onset of apoptosis?

As shown in Figure 6g, there is a significantly lower level of putrescine in the brainstem following combination treatment compared with treatment with DFMO alone. While this difference appears small, this is in part due to the scale of the y axis and in fact the level is 50% lower in combination treated mice compared with DFMO treated mice (Supplementary Figure 22). It is also important to note that the level of putrescine in the vehicle treated brain is almost double that of normal brain, as shown in Figure 1f. Thus, the level in combination treated tumours is significantly lower than in normal brain and likely has passed a critical threshold leading to the profound impact on survival. As requested by Reviewer 3 we have also now included the spermidine:spermine ratios which are significantly lower in the combination treated animals (Figure 6h).

As suggested, we have further examined the effects of this profound polyamine depletion *in vivo*. It is well established that DFMO is an irreversible inhibitor of ODC1 thus reducing polyamine synthesis. Eukaryotic translation factor 5A (eIF5A) is a key protein regulating several processes during translation, which requires a unique post-translational modification for its activity. The hypusination of eIF5A requires the activity of deoxyhypusine synthase (DHPS) and deoxyhypusine hydroxylase (DOHH) both of which require spermidine as a substrate (Nakanishi

& Cleveland, 2016, *Amino Acids*, 48: 2353-2362). It has been shown that inhibition of ODC1 either by DFMO or knockdown leads to reduced hypusination of eIF5A and subsequently decreased protein translation (Zhang et al, 2019, *Molecular Cell*, 76: 110-125). A key target of hypusinated eIF5A is Lin28B/Let7 with DFMO shown to affect it in neuroblastoma (Zhang et al, 2019, *Molecular Cell*, 76: 110-125). Furthermore, in a separate study the combination of DFMO with CGC7, a DHPS inhibitor, also was shown to reduce eIF5A hypusination and cell proliferation and increase apoptosis in neuroblastoma cells (Schultz et al, 2018, *Biochemical Journal*, 475: 531-545). As suggested by the reviewer we evaluated by gene expression analysis for potential effects on the Lin28B/Let7 axis in single agent and combination treated DIPG cells. We observed a significant decrease of Lin28B mRNA levels in DFMO-treated cells. Interestingly AMXT 1501 lead to an increase in Lin28B indicating that despite decreased polyamine uptake, increased polyamine synthesis leads to upregulation of Lin28B. The combination of both polyamine inhibitors led to a reduction in the Lin28B expression to similar levels as DFMO-only treatment. Subsequent analysis revealed enhanced Let7 miRNA levels following DFMO and combination treated DIPG cells. These findings are included in Supplementary Figure 26a and in the results section (page 10, paragraph 2, lines 231-237). We also evaluated the effect on mTORC1, another pathway that regulates protein translation. We observed significantly lower levels of phosphorylated mTOR and phosphorylated 4EBP1, a downstream target, which is directly involved in protein translation. These findings are included in Supplementary Figure 26b and at the results section (page 11, paragraph 1, lines 237-239). Our data suggest that protein translation pathways are affected by DFMO and AMXT 1501 combination treatment and the reduced expression of ODC1 mRNA could be due to effects on translational machinery.

10. The survival curves are impressive – however analysis requires that a multiple comparisons test be applied to the log-rank test. This can be done in GraphPad using “compare a stack of p-values”. After such a multiple comparisons test was undertaken, I highly doubt that DFMO would be found to be statistically significantly extending survival compared to control. The discontinuous X axis employed in figure 4a should not be used, since this is specifically done by the authors to try to highlight the non-clinically significant difference in survival between DFMO and control in SUDIPGVI.

As suggested, we have analysed the survival curves using a multiple comparisons test. DFMO treatments were found to significantly enhance survival comparing to controls. Analysis is included as a supplementary tables 2, 3 and 4. In addition statistical analysis has been amended in methods (page 21, paragraph 2, lines 494-498), results section (page 8, paragraph 1 lines 183-184 and lines 185-186 and page 9, paragraph 1, lines 202-203) and legend for Figure 6 (page 31, paragraph 1, lines 758-759). In addition, the x-axis for the survival curve in Figure 6a has been changed to a continuous line as suggested.

11. Supplemental figure 6 should be repeated in multiple DIPG cell lines to add robustness to the author’s assertion that DFMO treatment decreases ODC1 protein expression and AMXT1501 increases ODC1 protein expression

This is now supplemental Figure 11. We have repeated the immunoblotting analysis in two additional primary cultures, RA055 and SU-DIPGVI. Similar to HSJD-DIPG007 cells (Supplementary Figure 11a,b,c) we have observed a significant increase in ODC1 levels following AMXT 1501 treatment in SU-DIPGVI and RA055 DIPG cells. The difference in ODC1 level following treatment with DFMO alone did not reach significance, however the addition of DFMO

to AMXT 1501 treatment led to a significant decline in ODC1 protein levels. Overall, combination treatment appeared to reduce ODC1 protein back to similar levels as seen with DFMO treatment alone in all 3 primary DIPG cultures. These results are included in Supplementary Figure 11b (RA055) and 11c (SU-DIPGVI) and in the results section (page 6, paragraph 2, lines 137-139).

12. MRC5 and their normal human astrocytes (NHA) should be defined better in the methods and described in the figure legends.

We have now included in the methods section the information requested for human lung fibroblast cells (MRC5) on page 14, paragraph 2, lines 327-329. In addition, we have amended the legends for Figures 1 and 2 to include the full names and abbreviations for MRC5 cells and normal healthy astrocytes (NHA). This information can be found for Figure 1 legend on page 27, paragraph 1, lines 664-666.

13. The authors make claims about ornithine pathway inhibition in pediatric non-midline GBM – however they do not show data of activity in this manuscript that I can see (there are some supplemental figures showing expression of pathway genes). They should either include data showing similar activity of polyamine pathway inhibition in this paper or should remove those claims.

We agree and have removed this statement from the discussion section as suggested.

14. The authors refer to HSJD-007 as a “PDX” – this implies that it has never been grown in plastic and is serially passaged. I do not believe this is that case and the term “PDX” should be changed to “DIPG patient-derived cell line grown as an orthotopic xenograft”.

We thank the reviewer for this comment and have replaced the term PDX as suggested to “DIPG patient-derived cells grown as an orthotopic xenograft model”. These changes can be found in the Results section page 7-8, lines 166-169.

15. Please double check the stereotactic coordinates for injection since “coordinates: 0.5 mm anterior, 6.0mm lateral” seems reversed from the usual which would be just lateral to the midline and either 6 mm posterior from bregma or slightly anterior or posterior from lambda.

We have corrected the stereotactic coordinates in the Methods section (page 18, paragraph 2, line 420-422) as following, “0.5mm lateral to midline, 6mm posterior to bregma suture and 3.5mm deep”.

Reviewer 2

1. The article would benefit from separation in clear categories (abstract, introduction, results), each with appropriate subheadings. Similarly, the methods should be organized more clearly.

We have now adapted the manuscript according to Nature Communications specifications. The manuscript now includes an Abstract (page 2), Introduction (pages 2-4), Results (pages 4-11), Discussion (pages 11-14), and Methods (pages 14-21), sections.

2. The article should be proofread for clarity; punctuation should be corrected.

The manuscript has been thoroughly proofread and all errors corrected.

3. With molecular evidence mounting, DIPG is now a sub-entity of diffuse midline gliomas. This study would benefit from a discussion and demonstration on how its results could be applicable to other midline gliomas (e.g. thalamic).

We thank the reviewer for this comment and have evaluated the cytotoxic efficacy of both polyamine inhibitors in two primary thalamic brain tumour cultures AUS-DIPG-017 (H3K27wt) and P001805 (H3K27M). Using alamar blue assays we found that the combination of DFMO with AMXT 1501 synergistically decreased cell proliferation in both primary cultures (Supplementary Figure 13 c and d). These results correlate closely with those seen in DIPG cultures HSJD-DIPG007, SU-DIPGVI and SU-DIPGXVII. The cytotoxic efficacy curves are included in the results section (page 7, paragraph 1, lines 148-152). We have now discussed how the results could be applicable to other midline gliomas on page 12, paragraph 2, lines 274-277. Further information on the molecular status of both primary cultures has been included in Supplementary Table 1.

4. The authors did not discuss the histone 3 status of their cell lines and mouse models. The h3.3k27m mutation has been shown to pertain to worse prognosis in DIPG with treatments against this genetic subtype being developed.

We have included mutational status as well as additional information on tissue type and patient treatment in Supplementary Table 1.

5. Similarly, the authors did not consider DIPG heterogeneity and molecular profiles and how these could affect response to therapy. This paper (10.1016/j.ccell.2017.08.017) should be referenced.

We have included a comment addressing DIPG heterogeneity and molecular status according to Mackay et al in the discussion section (page 12, paragraph 2, line 274-277).

6. The authors briefly touch on BBB permeability. However, they do not demonstrate how the BBB is intact in their animal models nor do they discuss whether DFMO and AMXT 1501 are, in fact, BBB permeable.

We agree with the reviewer and note that while SU-DIPGVI, HSJD-DIPG007 and other orthotopic DIPG models have been used to test drug efficacy in multiple publications the integrity of the BBB has not been formally published to date. We have now, for the first time, been able to show that the BBB remains intact in these orthotopic animal models, using the Evans Blue (EB) extravasation method (Radu & Chernoff, 2013, J Vis Exp, e50062; Goldim et al, 2019, Curr Protoc Immunol, 126:e83). Briefly EB dye was administered intravenously to normal Balbc/Nude animals, matrigel-injected animals and animals intracranially injected with RA055, SU-DIPGVI and HSJD-DIPG007 cells. Within 30 min of EB administration the blue dye visibly stained the skin of all the animals. Subsequently myocardial perfusion was performed, and brains, and spleen were analysed for EB levels with a spectrophotometer. The results showed that uninjected, matrigel-injected and DIPG-injected animals did not display any EB levels in brainstem and

cortical regions, consistent with an intact BBB. In contrast the spleen displayed high levels of EB extravasation into the tissue. These results are included in Figure 5 and results section (page 8, paragraph 1, lines 170-177). In addition, a detailed protocol is included in methods section (page 19, paragraph , lines 439-450).

7. Most control graphs lack error bars. The authors state that all controls were done in triplicates. However, if that is the case control graphs would still require error bars. We recommend enquiring with a statistician on how to best represent these data (normalize each value against the average of the control, and then normalize, rather than normalizing average against average).

We thank the reviewer for this comment and have corrected the control bar graphs to contain error bars for Figures 1d, 1e, Figures 2e, 2f and 2g. In addition, we have included error bars in the controls for the Supplementary Figures Suppl 6b, Suppl 7, Suppl 10a, 10b and Suppl 27a,27b.

8. The authors mention in passing that their treatment is safe. However, they fail to show so. In clinical trials, DFMO treatment has been shown to have Grade 3 toxicities in children with neuroblastoma, with transaminitis and neutropenia being common side effects. The authors should further test for immune system-related toxicities in immune-competent mice.

DFMO has been used for several decades in paediatrics, and is established as having minimal toxicity, even at the very high doses intravenously in children for treatment of African sleeping sickness. The most common toxicity is hearing loss which is reversible following withdrawal of treatment (Meyskens & Gerner, Clin Cancer Res. 1999). Transaminitis and neutropenia are uncommon and were only seen occasionally in heavily pretreated children treated for neuroblastoma when administered in combination with cytotoxic chemotherapy (Marachelian et al, JCO, 2018 36:15_suppl, 10558-10558). In combination with chemotherapy the recommended phase 2 has been established as 6750 mg/m²/day – a very high dose compared with other anti-cancer compounds – and was tolerated on a daily schedule for up to a year of treatment (Marachelian et al, JCO, 2018 36:15). We have now noted this in the discussion on page 11, paragraph 2, lines 255-259.

9. The toxicity profile of AMXT-1501 is similarly poorly characterized and the field would benefit from an in-depth safety assessment in mice (at least).

We have now included additional data to provide more detail on the toxicity and tolerability of DFMO in combination with AMXT 1501. In Supplementary Figure 21 we have shown the average weights of the animals for the entire duration of the treatments for animal models SU-DIPG-VI-LUC (a) and HSJD-DIPG007 (b). In both DIPG models DFMO and AMXT 1501 single agent treated cohorts displayed similar body weights to vehicle treated animals with the exception of the combination treated animals, which actually exhibited an increase in weight. These findings are included in the results section (page 9, paragraph 1, lines 203-206).

In addition, we performed biochemical analysis of mice treated with AMXT 1501 at higher doses 5mg, 7.5mg and 10mg/kg/day (see Table below). There was no change in any of the parameters as a single agent or in combination with DFMO. Importantly, a much lower dose of 2.5 mg/kg was used for the treatment studies therefore we have not included these results in the manuscript. While extensive toxicology studies are beyond the scope of this manuscript, we

note that our collaborators at Aminex have performed comprehensive toxicological analysis in CD1 animals using 3 different doses of AMXT 1501 (2.5, 5 and 10 mg/kg/day) given continuously for 28 days. During dosing and 14 days following treatment no changes were observed in body weights, food consumption or ophthalmology assessments. AMXT1501 in combination with DFMO is currently being tested in an adult Phase 1 clinical trial, with no drug related severe adverse events reported to date.

10. Figure 1a and 1b. The authors are unclear on the origin of these samples. Were these DIPG samples obtained from a consortium? It is similarly unclear if 'signal intensity' refers to RNA or protein expression. Further, it needs to be clarified what 'normal brain' refers to: adult brain samples or age-matched pediatric brainstem (how and why was it obtained?)? Further, the use of 35 samples only, in light of papers such as the Mackay discussed above, where ODC1 and SAT1 levels could be obtained (at least on the mRNA level) is rather underwhelming. ODC1 and SAT1 levels should also be compared with other known DIPG mutations and drivers.

The data originally used in Figures 1a/b were obtained from Buczkowicz et al Nature Genetics 2014, 46:451-46. The authors of that manuscript used autopsy derived patient-matched frontal lobe or occipital lobe as normal brain. We have since updated these figures to include analysis based on a larger set of data (11 foetal normal brain and 49 DIPG/DMG samples) obtained from the Zero Childhood Cancer program and McGill University. Similar to the original figures this new analysis indicates significantly higher expression levels of ODC1 and reduced levels of SAT1. These findings are included in the results section (page 4, paragraph 2, 81-84 and lines 87-90). We could not use the Mackay cohort for this analysis as they do not include healthy brain as a control. However, we have been able to compare the expression levels of ODC1 in H3wt and H3K27M DIPG tumours in that cohort. We found there was no statistically significant difference in the expression levels of ODC1 between H3wt, H3.1K27M and H3.3K27M (Supplementary Figure 3), indicating that ODC1 expression is independent of the molecular status of DIPG. This result is included in the results section, page 4, paragraph 2, lines 86-87.

11. Figure 1c-1e. The changes in ODC1 and SAT1 are not consistent across cell lines, with some cell lines not differing from control. Further, using MRC5 as a control is controversial; pediatric brainstem tissue (astrocytes) would be a better control.

We have performed densitometry in the western blot bands and have confirmed significantly higher levels of ODC1 in 4 DIPG cultures compared to NHA cells (Supplementary Figure 4). In addition, we repeated the gene expression analysis for ODC1 and SAT1 and have now included NHAs as well as 2 additional normal paediatric astrocyte cultures. We found significantly higher expression of ODC1 and concomitant reduced levels of SAT1 in DIPG cultures compared to each healthy astrocyte culture (Figure 1d,e). These data are included in the results section (page 4, paragraph 2, lines 90-93).

12. Figure 2a. Again, the use of 'normal brain' is controversial and should be explained.

The data used in Figure 2a are obtained from Buczkowicz et al Nature Genetics 2014, 46:451-46. The authors have used autopsy derived patient-matched frontal lobe or occipital lobe as normal brain. This result has been replaced with new analysis based on a larger set of data (11 foetal normal brain and 49 DIPG) obtained from the Zero Childhood Cancer precision medicine

platform and McGill University as described above (Question 10). The expression levels of SLC3A2 are significantly higher in DIPG compared to normal brain.

13. Figure 2d. The WB is underwhelming and, by eye, hardly significant. A quantification graph is necessary.

We agree and have now performed densitometry and have presented the quantification results as a bar graph. This is provided as supplementary information (Supplementary Figure 9). We confirmed that the protein levels of SLC3A2 significantly increase upon treatment with DFMO ($P < 0.005$).

14. Figure 2e. The significance of this graph is questionable. The authors show an increase in mRNA expression 12 and 24 hours after treatment. All cell experiments thus far, and all experiments following, have an incubation time of 3 days. This timepoint should be used. Further, the statistics used seem to be incorrect (ANOVA instead of t-test).

Statistical analysis using one-way ANOVA confirms that SLC3A2 gene expression significantly increased at 12h and 24h timepoints. We have also detected significantly higher mRNA expression levels at 48 and 72h timepoints. We have included these results in Figure 2e and the results section (page 6, paragraph 2, lines 125-126).

15. Figure 2g. The authors used the wrong statistical analysis tool; this graph would benefit from a one-way ANOVA with multiple comparisons to show how increased drug concentrations lead to decreased uptake.

We have analysed the data with one-way ANOVA and confirmed that the uptake of radiolabeled spermidine is significantly reduced in comparison to control and the lowest concentration of AMXT 1501. Statistical analysis is included in the figure 2 legend.

16. Supplementary Figure 7. This figure somewhat undermines the authors' point, as combination treatment is not always superior to each drug alone and, judging from the graphs, not even additive. The authors should repeat these experiments and address their conclusions in light of these findings.

We thank the reviewer for this comment and have now included in Supplementary Figure 12 synergy scores. All polyamine concentrations levels were found to be synergistically reduced in both primary cultures following combination treatment, with the exception of putrescine where the effect was additive in SU-DIPGVI cells treated with the combination.

17. Figure 3a. The graph of DFMO is not concordant with the one shown in figure 1h, as the latter graph had an end survival of 20% at the highest concentration. The one in 3a does not.

This figure is now Figure 4a. We agree and note that there is only a minor variability between these experiments, each of which were repeated in triplicate.

18. Figure 3a-c. Combination studies are correctly executed across an array of concentrations. CI's, however, are also dependent on, and should be indicated at, different drug

concentrations. This is the case because, at extremes of concentration, combinatorial effects might wane. As such, the relevance of the combinatorial indices shown is unclear.

We have included the individual CI values for each drug concentration tested for graphs 4a and 4b as well as Supplementary Figures 13a and 13b. These are noted on each graph and show synergy at almost every concentration, apart from the extremes.

19. Figure 3e. This WB needs quantification as the differences across treatments are not convincing.

We have performed densitometry for Figure 4e and have included the results as a bar graph. The results show that treatment with DFMO and AMXT 1501 leads to significantly higher levels of caspase 8 and cleaved parp compared to single agents. These results are included in Supplementary Figure 17. In addition, we have also confirmed these findings on additional primary culture SU-DIPGVI cell (Supplementary Figure 16b). These results are included on page 7, paragraph 1, lines 159-161.

20. Page 6. The authors indicate that their mouse models maintain an intact BBB and cite reference 30. Of note, Hennika et al. used a different mouse model from those used in this paper which, nonetheless, had a permeable BBB. Further, it remains the matter of debate whether xenograft mouse models can maintain an intact BBB.

As discussed above (question 6) we have now shown that the BBB is intact in the models used in these experiments.

21. Figure 4a, 4d, 4f. These graphs would benefit by indicating the median survival for each treatment group. Further, the authors should discuss the significance of the fact that DFMO treatment alone yield a significant survival benefit compared to control. The timing of treatment is also unclear and should be better explained in the methods.

We would like to note that Figure 4 is now Figure 6. We have included the median survival for each treatment cohort in the figure legends for Figures 6a, 6d and 6f. In order to indicate the start and completion of the treatment we placed grey shaded areas in the survival curves. In addition, we have included the starting days for the treatments in the methods section (page 18, paragraph 2, lines 420-422).

22. Figure 4b and 4c. These figures lack the single drug control. Further, it would be beneficial to know how the luminescence signal progressed from implantation to when the authors start showing changes.

We have now included in Figure 6c the luminescence graphs for DFMO and AMXT 1501 treated animals. The first luminescence image we obtained was at day 56 post intracranial injection of DIPG cells therefore we are not able to include previous images to show luminescence progression following implantation.

23. Figure 4g. these comparisons require one-way ANOVA with multiple comparisons rather than repeated t-tests. Further, the authors looked at levels of putrescine alone as a measure of pathway inhibition – the changes in spermidine and spermine are not impressive (if

existing). A more in-depth characterization of the in vivo levels of these agents is necessary. Further, it would be beneficial to know the level of ODC1 and SLC3A2 following treatment in vivo.

We thank the reviewer for this comment and have reanalysed our data using one-way ANOVA with multiple comparisons. We can confirm that the putrescine levels are significantly reduced between single agents and combination treatment (DFMO vs Combination $p=$ and AMXT 1501 vs combination $p<0.01$ and $p<0.001$ respectively). We note that the labelling of this figure has been changed to Figure 6g. Furthermore, we originally provided in Figure 6h effects of polyamine inhibitors on spermidine and spermine levels in brain tissue samples treated with single agents and combination. This figure has been changed to include the spermidine to spermine (spd:spm) ratio as requested by the Reviewer 3. We observed significantly lower spd:spm ratio in combination treated animals compared to AMXT 1501 and vehicle treated animals indicating a reduction in the polyamine biosynthetic pathway. These findings are included in the results section (page 9, paragraph 2, lines 213-217). In addition, as suggested by the reviewer we used western blots to evaluate SLC3A2 protein levels in brain tissue isolated from SU-DIPGVI-LUC xenografted animals treated with single agents and combination for 1 week. These results are included in Supplementary Figure 23 and show a significant increase in SLC3A2 protein levels following DFMO treatment. These findings are included in the results section page 10, paragraph 1, lines 224-226.

Reviewer 3

1. In figure 1f and 4g, the authors measure putrescine levels, but really need to show spm and spd in the main figures. The spd:spm ratio may also be informative. This perhaps would lead to additional experiments that would provide novelty in terms of mechanism. For example, looking at the polyamine levels in Fig 4g and supplemental fig 12a and 12b, there seems to be minimal added change in polyamines when AMXT 1501 is added to DFMO, despite dramatic differences in survival and tumor growth. How to the authors explain this? It may be worthwhile asking what is happening to the PTEN and MTOR pathway. AMD1 activity levels, dcSAM levels, and SMOX and PAOX activity levels? Is there an accumulation of H₂O₂ or indicators of oxidative stress?

We thank the reviewer for this comment and have now included the spermidine to spermine ratios (spd:spm) in Figures 1g and 6h. Figure 1g shows significantly higher levels of spd:spm in the brains of DIPG-engrafted animals comparing to tumour-free animals suggesting increased polyamine metabolism in the presence of a DIPG tumour. This finding is included in the results section (page 5, paragraph 1, lines 97-100). Figure 6h shows significantly lower levels of spd:spm in animals treated with dual polyamine inhibitors. It is important to note that polyamine levels presented here were determined after one week of treatment rather than following treatment completion or at endpoint where we observe significant changes in survival and tumour growth. This finding is included in the results section (page 9, paragraph 2, lines 224-226).

Given the role of polyamines in many biological processes we have looked as suggested for effects on the mTOR pathway. We found following 24h of treatments the combination of DFMO/AMXT 1501 reduced the phosphorylation levels of mTOR (P-2448) and downstream target 4EBP1 indicating a complex integration of oncogenic pathways and metabolic programs (Supplementary Figure 26b). This result is included in the results section (page 10-11, paragraph

1, lines 243-245). Polyamine synthesis can also be driven by methionine metabolism a pathway also influenced by mTORC1 signalling as recently shown in prostate cancer (Zabala-Letona et al, 2017, Nature 547:109-113). Furthermore, another research group recently evaluated the integration of the methionine pathway and polyamine metabolism and exploited therapeutically this metabolic vulnerability in prostate cancer (Affronti et al, 2020, Nature Communications 11:52-66). In the methionine cycle, adenosylmethionine decarboxylase 1 (AMD1) drives the conversion of s-adenosylmethionine (SAM) to decarboxylated SAM a metabolite, which is subsequently involved in spermidine and spermine synthesis (Affronti et al, 2020). We do not know if in DFMO/AMXT 1501 treatments affect the activity of AMD1, however given the putrescine to spermidine (put:spd) and spd:spm levels are reduced, it could be possible that dcSAM levels that contribute to the synthesis of these polyamines are low as well. We have included the put:spd levels as a Supplementary Figure 22 and in the results section (page 9, paragraph 2, lines 213-217). In addition in the discussion we have noted that a limitation of our study is that we have not been able to evaluate the potential implication of the methionine cycle in the regulation of the polyamine pathway, and the effect of DFMO/AMXT 1501 (page 13, paragraph 2). Polyamines are known to influence oxidative metabolism through acting like scavengers for reactive oxygen species (Murray-Stewart et al 2018, J Biol Chem. 293: 18736-18745). During the retro-conversion of spermine to spermidine there is elevated production of H₂O₂. As suggested above our results indicate low put:spd and spd:spm ratios upon DFMO/AMXT 1501 combination treatment suggesting it is less likely there is a back conversion to putrescine due to increased activity of SMOX and PAOX.

2. It is not clear from the methods if there is serum in the media used for studies where polyamines are added to the media to affect growth and migration. If so, is aminoguanidine added to prevent amine oxidase activity.

Proliferation assays were performed in serum free media however for migration assays foetal calf serum was added. We have included this information in the Methods section (page 17, paragraph 3, lines 409). We did not add aminoguanidine.

3. It might be informative to ask if the IC₅₀s defined in Fig 1h correlate with quantification of ODC1 levels by Western in fig 1c.

We thank the reviewer for this suggestion and have now performed densitometric analysis on the ODC1 protein levels presented in Figure 1c. We have observed a correlation between IC₅₀s and ODC1 protein levels. This analysis is presented as lines of best fit in Supplementary Figure 5 and this finding is included in the results section (page 4, paragraph 2, lines 93-94).

4. What are the sample sizes in fig 2b and 2c?

We have included sample sizes in the methods section (page 16, paragraph 2, lines 376-378) and in the figure legend 2 (page 28, paragraph 2, lines 702-705). This analysis includes 28 DIPG/DMG samples, 148 other high-grade paediatric cancers and 17 high-risk neuroblastoma samples.

5. The Western blot in Fig 2d, meant to demonstrate upregulation of SLC3A2, is not so convincing. Conceptually, it is not important that there is an increase in SLC3A2. It is sufficient to say that it is still there, and therefore targetable. Showing convincingly that it is upregulated after DFMO treatment really does not matter much. One does not need to

upregulate the transporter for it to be more used in the context of DFMO treatment. The key point comes in Fig 2f demonstrating increased uptake, and this is true regardless of whether or not the Western convincingly demonstrates upregulation of the transporter. There is, however, an important missing experiment: If the argument is that DFMO induced increases in uptake help to make cells resistant to DFMO, then if spd/spm are added to the media does this make the cells more resistant to DFMO, and can this be countered by AMXT 1501?

We thank the reviewer for this comment and as requested we have performed densitometry for Figure 2d. We can confirm that the protein levels of transporter SLC3A2 are significantly higher upon treatment with DFMO, which are in agreement with the spermidine uptake results presented in Figure 2f. The densitometry results are included in Supplementary Figure 9. In addition, we have evaluated whether an increase in polyamine uptake would make DIPG cells resistant to DFMO and subsequently sensitive to polyamine transporter inhibitor. We performed alamar blue assays with and without the addition of polyamines and measured the response to DFMO and combination treatment. As predicted, the addition of polyamines rendered DIPG cells less sensitive to DFMO treatment. However, following the treatment with AMXT 1501, the addition of exogenous polyamines had no effect on cell survival (Supplementary Figure 14). These results indicate that AMXT 1501 is able to counter the resistance mediated during DFMO treatment. These findings are included in the results section (page 7, paragraph 1, lines 154-156).

6. Supplemental Fig 7 contains important data that should be included in the main figures for at least one of the models. Going back to point 1, it is interesting that all three polyamines are reduced in these culture models, but it looks very different in vivo. Can the authors explain this?

We thank the reviewer for this comment and have now included the SU-DIPGVI results as main Figure 3 and the results for HSJD-DIPG 007 as Supplementary Figure 12. We would argue that the *in vitro* results in Figure 6g and the *in vivo* results in Supplementary Figure 12 are quite similar with significantly lower putrescine levels in combination treated samples in comparison to cells treated with either single agent. Spermidine and spermine levels are also significantly reduced in SU-DIPG-VI cells treated with combination (Figure 3). Similarly, the *in vivo* data show a significantly lower spermidine to spermine ratio comparing with AMXT1501 treated animals (Figure 6g).

7. Based on data from Supplemental figure 11, the authors argue that these drugs can cross the blood-brain barrier. It would seem that an important control would be to include animals that have not had anything implanted in their brainstem. This would control for the possibility that the physical process of implantation, or the presence of foreign tumor, influences the ability of the two drugs to cross the blood-brain barrier.

We agree with the reviewer and for this reason we have included the levels of both AMXT 1501 and DFMO in the frontal brain region, the area of the brain which was not implanted with DIPG cells. The DFMO and AMXT 1501 concentrations in the frontal lobe is the same as in the DIPG-engrafted brainstem region. In addition to this, as requested by reviewer 2 (Question 6), we have examined the BBB permeability of Balbc/Nude animals uninjected, matrigel injected and DIPG injected and have observed that implantation of the tumours or injection of matrigel does

not influence extravasation of Evans Blue dye indicating the presence of an intact BBB (Figure 5). These additional findings are included in the results section (page 8, paragraph 1, lines 170-177).

8. Though toxicity of the combination was minimal, how was toxicity measured, what was the toxicity, at what dose, given for how long?

We have addressed this comment above for Reviewer 2 (Question 9).

9. The authors state that “Our results suggest that DIPG tumors are critically dependent on elevated polyamine pathway activity, with tumour growth driven by the addition of polyamines...” It was not demonstrated that tumor growth was driven by the addition of polyamines. Fig 1g shows some increased proliferation in culture, but that is very different that the quoted statement.

We thank the reviewer for this comment and have rephrased our statement as “Our results suggest that DIPG tumours are critically dependent on elevated polyamine pathway activity, with DIPG cell proliferation driven by the addition of polyamines” (page 11, paragraph 2, lines 260-262).

REVIEWERS' COMMENTS:

Reviewer #1 (Remarks to the Author):

The authors have satisfied my concerns and the manuscript is significantly strengthened and the work makes a major contribution to the field.

Minor issues: Figure 1g - I believe the x-axis label should read "DIPG tumorcells injected" similar to 1F and not "vehicle" as written. It might be better labeled "HSJD007 DIPG tumor cells injected" to help the reader better understand the data presented.

In many cases in the figures, the error bars are listed as SEM. Some journals do not allow this - it is a tool to make the error bars smaller since SEM by definition is smaller than SD. The best practice as graphpad states: "If you are plotting on a column graph fewer than 100 or so values per data set, create a scatter plot that shows every value. What better way to show the variation among values than to show every value?"

Reviewer #3 (Remarks to the Author):

In the revised article by Khan et al, entitled "Dual Targeting of polyamine synthesis and uptake in diffuse intrinsic pontine gliomas", the authors have adequately addressed my concerns. The addition of hypusine related and MTOR related data help to strengthen mechanistic insights. The inclusion of spd:spm ratios strengthen the observed effects of DFMO and AMXT 1501. The authors make a reasonable argument that PAOX and SMOX are failing to compensate, though I would be curious about the dynamics. Not that it is needed in this manuscript, but it would be interesting to know if early in treatment there are spikes in SMOX and/or PAOX activity in an attempt to compensate for sudden reduction of intracellular polyamines.

The new studies in Supplemental Figure 14 are very important and strengthen the idea that the efficacy of AMXT 1501 is owed to its ability to block import of exogenous polyamines as a means of mitigating the block in synthesis.

The authors have addressed the BBB issue in what appears to be a reasonable way. However, this reviewer confesses to not being an expert on this topic and will gladly defer to reviewer 2 who raises this issue in point #6.

Title: Dual targeting of polyamine synthesis and uptake in diffuse intrinsic pontine gliomas

We thank the editorial team and the referees for their thorough review of the revised manuscript. We have addressed the comments from reviewers 1 and 3. The specific changes and responses are addressed below.

Reviewer #1:

Minor issues: Figure 1g - I believe the x-axis label should read "DIPG tumorcells injected" similar to 1F and not "vehicle" as written. It might be better labeled "HSJD007 DIPG tumor cells injected" to help the reader better understand the data presented.

We thank the reviewer for this comment and have now corrected the x-axis label for Figure 1g.

In many cases in the figures, the error bars are listed as SEM. Some journals do not allow this - it is a tool to make the error bars smaller since SEM by definition is smaller than SD. The best practice as graphpad states: "If you are plotting on a column graph fewer than 100 or so values per data set, create a scatter plot that shows every value. What better way to show the variation among values than to show every value?"

As requested by the reviewer we have included scatter plots within the bar graphs for the main figures and supplementary figures. This has also been addressed in the editorial checklist

Reviewer #3:

In the revised article by Khan et al, entitled "Dual Targeting of polyamine synthesis and uptake in diffuse intrinsic pontine gliomas", the authors have adequately addressed my concerns. The addition of hypusine related and MTOR related data help to strengthen mechanistic insights. The inclusion of spd:spm ratios strengthen the observed effects of DFMO and AMXT 1501. The authors make a reasonable argument that PAOX and SMOX are failing to compensate, though I would be curious about the dynamics. Not that it is needed in this manuscript, but it would be interesting to know if early in treatment there are spikes in SMOX and/or PAOX activity in an attempt to compensate for sudden reduction of intracellular polyamines.

We thank the reviewer for his comment and agree that it will be of great interest to determine the SMOX and PAOX activity. We intend to address this in future studies as we will be continuing our investigations in other aggressive paediatric models.

The authors have addressed the BBB issue in what appears to be a reasonable way. However, this reviewer confesses to not being an expert on this topic and will gladly defer to reviewer 2 who raises this issue in point #6.

We thank the reviewer for this comment and believe we have addressed the BBB permeability issues raised by reviewers 2 and 3 in the previously revised manuscript.